



GTS v1.0: A Macrophysics Scheme for Climate Models Based on a Probability
Density Function
Chein-Jung Shiu[1,*], Yi-Chi Wang[1], Huang-Hsiung Hsu[1], Wei-Ting Chen[2], Hua-Lu
Pan[3], Ruiyu Sun[4], Yi-Hsuan Chen[5], and Cheng-An Chen[1]
Research Center for Environmental Changes, Academia Sinica, Taiwan
Department of Atmospheric Sciences, National Taiwan University, Taiwan
Retired Senior Scientist, National Centers for Environmental Prediction, NOAA,
USA
National Centers for Environmental Prediction, NOAA, USA
Department of Climate and Space Sciences and Engineering, University of
Michigan, USA

24 Correspondence: Chein-Jung Shiu (email: cjshiu@rcec.sinica.edu.tw)





Short title: Macrophysics for Climate Models
Key points:
1) A cloud macrophysics scheme utilizing grid-mean hydrometeor information is
developed and evaluated for climate models.
2) The GFS-TaiESM-Sundqvist (GTS) scheme can simulate variations of cloud fraction
associated with relative humidity (RH) in a more consistent way than the default
scheme of CAM5.3.
3) Through a better cloud–RH relationship, the GTS scheme helps to better represent
cloud fraction, cloud radiative forcing, and thermodynamic-related climatic fields in
climate simulations.





Abstract
Cloud macrophysics schemes are unique parameterizations for general circulation
models. We propose an approach based on a probability density function (PDF) that
utilizes cloud condensates and saturation ratios to replace the assumption of critical
relative humidity (RH). We test this approach, called the GFS-TaiESM-Sundqvist (GTS)
scheme, using the macrophysics scheme within the Community Atmospheric Model
version 5.3 (CAM5.3) framework. Via single-column model results, the new approach
reveals a stronger linear relationship between the cloud fraction (CF) and RH when
compared to that of the default CAM5.3 scheme. We also validate the impact of the
GTS scheme on global climate simulations with satellite observations. The simulated
CF is comparable to CloudSat/CALIPSO data. Comparisons of the vertical
distributions of CF and cloud water content (CWC), as functions of large-scale dynamic
and thermodynamic parameters, with the CloudSat/CALIPSO data suggest that the
GTS scheme can closely simulate observations. This is particularly noticeable for
thermodynamic parameters, such as RH, upper-tropospheric temperature, and total
precipitable water, implying that our scheme can simulate variation in CF associated
with RH more reliably than the default scheme. Changes in CF and CWC would affect
climatic fields and large-scale circulation via cloud–radiation interactions. Both
climatological means and annual cycles of many of the GTS-simulated variables are
improved compared with the default scheme, particularly with respect to water vapor
and RH fields. Different PDF shapes in the GTS scheme also significantly affect global
simulations.





1. Introduction

Global weather and climate models commonly use cloud macrophysics
parameterization to calculate the sub-grid cloud fraction (CF) and/or large-scale cloud
condensate, as well as cloud overlap, which is required in cloud microphysics and
radiation schemes [Slingo, 1987; Sundqvist, 1988; Sundqvist et al., 1989; Smith, 1990;
Tiedtke, 1993; Xu and Randall, 1996; Rasch and Kristjansson, 1998; Jakob and Klein,
2000; Tompkins, 2002; Zhang et al., 2003; Wilson et al., 2008a,b; Chabourea and
Bechtold, 2012; Park et al., 2014; Park et al., 2016]. The largest uncertainty in climate
prediction is associated with clouds and aerosols [Boucher et al., 2013]. The large
number of cloud-related parameterizations in general circulation models (GCM)
contributes to this uncertainty. In recent years, an increasing amount of research has
been devoted to unifying cloud-related parameterizations, for example by incorporating
the planetary boundary layer, shallow and/or deep convections, and stratiform cloud
(cloud macrophysics and/or microphysics) parameterizations, to improve cloud
simulations in large-scale global models [Bogenschutz et al., 2013; Park et al., 2014a,
2014b; Storer et al., 2015].
Some of these parameterizations use prognostic approaches to parameterize the CF
[Tiedtke, 1993; Tompkins, 2002; Wilson et al., 2008a, b; Park et al., 2016] while others
use diagnostic approaches [Sundqvist et al., 1989; Smith, 1990; Xu and Randall, 1996;
Zhang et al., 2003; Park et al., 2014]. Most of the diagnostic approaches used in GCM
cloud macrophysical schemes use the critical relative humidity threshold ($RH_c$) to
calculate CF [Slingo, 1987; Sundqvist et al., 1989; Roeckner et al., 1996]. In this type
of parameterization, GCMs frequently use the $RH_c$ value as a tunable parameter
[Mauritsen et al., 2012; Golaz et al., 2013; Hourdin et al., 2016]. There are some studies
on the verification of global simulations focused on the cloud macrophysical
parameterization [Hogan et al., 2009; Franklin et al., 2012; Qian et al., 2012;
Sotiropoulou et al., 2015]. In addition, many model development studies show the
impact of total water used in CF schemes on global simulations after modifying the $RH_c$
and/or the probability density function (PDF) [Donner et al., 2011; Neale et al., 2013;
Schmidt et al., 2014]. Some recent studies have attempted to constrain $RH_c$ from
regional sounding observations and/or satellite retrievals to improve regional and/or
global simulations [Quaas, 2012; Molod, 2012; Lin, 2014].
While many variations of the diagnostic Sundqvist CF scheme have been proposed,
most numerical weather prediction models and GCMs use the basic principle proposed
by Sundqvist et al. [1989]: the changes in cloud condensate in a grid box are derived
from the budget equation for RH. In the meantime, the amount of additional moisture
from other processes is divided between the cloudy portion and the clear portion





according to the proportion of clouds determined using an assumed $RH_c$. While changes
have been made to other parts of the Sundqvist scheme, the CF-$RH_c$ relationship still
applies in most Sundqvist-based schemes. As highlighted by Thompkins [2005], the
$RH_c$ value in the Sundqvist scheme can be related to the assumption of uniform
distribution for the total water in an unsaturated grid box such that the distribution width
(δc) of the situation when a cloud is about to form is given by:
$$\delta c = q_s(1 - RH_c), \tag{1}$$
where $q_s$ is the saturated mixing ratio.

We re-derived this equation by describing the change in the distribution width δ with
grid-mean cloud condensates and saturation ratio using the basic assumption of uniform
distribution from Sundqvist *et al*. [1989] rather than using the $RH_c$-derived δc, thereby
eliminating unnecessary use of the $RH_c$ while retaining the PDF assumption for the
entire scheme. This modified macrophysics scheme is named the GFS-TaiESM-
Sundqvist (GTS) scheme version 1.0 (GTS v1.0). It was first developed for the Global
Forecast System (GFS) model at the National Centers for Environmental Protection
(NCEP) and has been further improved for the Taiwan Earth System Model (TaiESM;
Lee *et al*., 2020) at the Research Center for Environmental Changes (RCEC), Academia
Sinica. Park *et al*. [2014] discussed a similar approach wherein a triangular PDF was
used to diagnose cloud liquid water as well as the cloud liquid fraction, and suggested
that the PDF width could be computed internally rather than specified, to consistently
diagnose both CF and cloud liquid water as in macrophysics. These authors also
mentioned that such stratus cloud macrophysics could be applied across any horizontal
and vertical resolution of a GCM grid, although they did not formally implement and
test this idea using their scheme. Building upon their ideas, we implemented and tested
this assumption with a triangular PDF in the GTS scheme.

In summary, this GTS scheme adopts Sundqvist's assumption regarding the partition
of cloudy and clear regions within a model grid box but uses a variable PDF width once
clouds are formed. It introduces a self-consistent diagnostic calculation of CF. Owing
to their use of an internally computed PDF width, GTS schemes are expected to be able
to better represent the relative variation of CF with RH in GCM grids.

A variety of assumptions regarding PDF shape can be adopted in diagnostic
approaches [Sommeria and Deardorff, 1977; Bougeault, 1982; Smith, 1990; Tompkins,
2002]. Some studies have investigated representing cloud condensate and water vapor
in a more statistically accurate way by using more complex types of PDF to represent
parameters such as total water, CF, and updraft vertical velocity [Larson, 2002; Golaz
*et al*., 2002; Firl, 2013; Bogenschutz *et al*., 2012; Bogenschutz and Krueger, 2013; Firl
and Randall, 2015]. In this study, we apply and investigate two simple and commonly
used PDF shapes—uniform and triangular—in our parameterization of the GTS



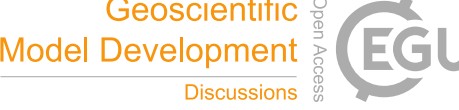

macrophysics scheme. Other complex types of PDF assumptions can also be used if
analytical solutions regarding the width of the PDF can be derived.

Most of the studies mentioned above estimate the CF via cloud liquid or total cloud
water. Earlier versions of GCMs used a Slingo-type approach to resolve the cloud ice
fraction [Slingo, 1987; Tompkins *et al*., 2007; Park *et al*., 2014]. On the other hand, the
current generation of global models participating in the Coupled Model
Intercomparison Project Phase 6 (CMIP6) have alternative approaches for the handling
of CFs associated with ice clouds. In the GTS scheme, the approach to cloud liquid-
water fraction parameterization is extended to the cloud ice fraction as well, wherein
the saturation-mixing ratio ($q_s$) with respect to water is replaced by $q_s$ with respect to
ice. This provides a consistent treatment for the cloud liquid and cloud ice fractions.
Many studies have argued that the assumption of rapid adjustment between water vapor
and cloud liquid water applied in GCM CF schemes cannot be applied to ice clouds
[Tompkins *et al*., 2007; Salzmann *et al*., 2010; Chosson *et al*., 2014]. In addition, it
would be difficult to represent the CF of mixed-phase clouds using such an assumption
[McCoy *et al*., 2016]. Applying a diagnostic approach to the cloud ice fraction similar
to that used for the cloud liquid fraction is indeed challenging and may result in a high
level of uncertainty. To investigate this issue, we also conduct a series of sensitivity
tests related to the super-saturation ratio assumption, which is applied when calculating
the cloud ice fraction in the GTS scheme.

2. Descriptions of scheme, model, and simulation setup

2.1 Scheme descriptions
Figure 1 illustrates the PDF-based scheme with a uniform PDF and a triangular PDF
of total water substance $q_t$. By assuming that the clear region is free of condensates and
that the cloudy region is fully saturated, the cloudy region ($b$) becomes the area where
$q_t$ is larger than the saturation value $q_s$ (shaded area). The PDF-based scheme
automatically retains consistency between CF and condensates because it is derived
from the same PDF. Here, we used the uniform PDF to demonstrate the relationship
between $RH_c$ and the width of the PDF. Using a derivation extended from Thompkins
[2005]:
$$b = \frac{1}{2\delta}(q_t + \delta - q_s). \qquad (2)$$

It is evident that, with the uniform PDF:
$$\delta_c = q_s(1 - RH_c). \qquad (3)$$

Therefore, $RH_c = 1 - \frac{\delta_c}{q_s}$. Thus, if the width $\delta$ of the uniform PDF is determined, then



$RH_c$ can be determined accordingly. This relation reveals that the $RH_c$ assumption of
the RH-based scheme actually assumes the width of the uniform PDF to be $\delta c$ from the
PDF-based scheme. As noticed by Thompkins [2005], the $RH_c$ used by Sundqvist *et al*.
[1989] for cloud generation can be linked to the statistical cloud scheme with a uniform
distribution. Building upon this finding, we eliminated the assumption of $RH_c$ by
determining the $P(q_t)$ with information about $q_V$ and $q_L$ provided by the base model.
In addition to the application of a PDF-based approach for liquid CF
parameterization, the GTS scheme also uses the same concept for parameterizing the
ice CF ($b_i$) as follows:
$$b_i = \frac{1}{2\delta}(\overline{q_I} + \overline{q_V} + \delta - sup * q_{si}),\tag{4}$$

where $\overline{q_I}$, $\overline{q_V}$, and $q_{si}$ denote the grid-mean cloud-ice mixing ratio, water-vapor mixing
ratio, and saturation mixing ratio over ice, respectively. In equation (4), $q_{si}$ is multiplied
by a supersaturation factor (*sup*) to account for the situation in which rapid saturation
adjustment is not reached for cloud ice. In the present version of the GTS scheme, *sup*
is temporarily assumed to be 1.0. Sensitivity tests regarding *sup* will be discussed in
Section 5.6.
A more complex PDF can be used for $P(q_t)$ instead of the uniform distribution in
our derivation. For example, the Community Atmospheric Model version 5.3 (CAM5.3)
macrophysics model adopts a triangular PDF instead of a uniform PDF to represent the
sub-grid distribution of the total water substance [Park *et al*., 2014]. Mathematically,
the triangular distribution is a more accurate approximation of the Gaussian distribution
than the uniform distribution and it may also be more realistic. Therefore, we followed
the same procedure to diagnose the CF by forming a triangular PDF with $\overline{q_L}$, $\overline{q_V}$, and
$\overline{q_s}$ provided. Moreover, by using a triangular PDF, we can obtain results that are more
comparable with the CAM5.3 macrophysics scheme because the same PDF was used.
By considering the PDF width, the CF ($b$) and liquid water content ($\overline{q_L}$) can be written
as follows:
$$b = \begin{cases} \frac{1}{2}(1 - s_s)^2 & if\ s_s > 0 \\ 1 - \frac{1}{2}(1 - s_s)^2 & if\ s_s < 0 \end{cases}\tag{5}$$

and:
$$\frac{q_L}{\delta} = \begin{cases} \frac{1}{6} - \frac{s_s^2}{2} + \frac{s_s^3}{3} - s_s b \\ \frac{1}{6} - \frac{1}{6}(3s_s^2 - 2s_s^3) - s_s b \end{cases},\tag{6}$$

respectively, where $s_s = \frac{q_s - \overline{q_t}}{\delta}$. From these two equations, we can derive the width of
the triangular PDF and calculate the CF ($b$) based on $q_s$, $q_t$, and $q_L$ instead of $RH_c$.


Notably, the PDF width for the total water substance can only be constrained when the
cloud exists. Therefore, the $RH_c$ is still required when clouds start to form from a clear
region.
In this study, GTS schemes utilizing two different PDF shape assumptions are
evaluated: uniform (hereafter, U_pdf) and triangular (hereafter, T_pdf). These two PDF
types are specifically formulated to evaluate the effects of the choice of PDF shape. A
triangular PDF is the default shape used for cloud macrophysics by the Community
Atmospheric Model version 5.3 (CAM5.3; hereafter, the Park scheme). The T_pdf of
the GTS scheme is numerically similar to that of the Park scheme except for using a
variable width for the triangular PDF once clouds are formed.

2.2 Model description and simulation setup
The GTS schemes described in this study were implemented into CAM5.3 in the
Community Earth System Model version 1.2.2 (CESM 1.2.2), which is developed and
maintained by DOE UCAR/NCAR. Physical parameterizations of CAM5.3 include
deep convection, shallow convection, macrophysics, aerosol activation, stratiform
microphysics, wet deposition of aerosols, radiation, a chemistry and aerosol module,
moist turbulence, dry deposition of aerosols, and dynamics. References for the
individual physical parameterizations can be found in the NCAR technical notes [Neale
*et al.*, 2010]. The master equations are solved on a vertical hybrid pressure−sigma
coordinate system (30 vertical levels) using the finite-volume dynamical core option of
CAM5.3.
We conducted both the single-column tests and stand-alone global-domain
simulations with CAM5.3 physics. The single-column setup provides the benefit of
understanding the responses of physical schemes under environmental forcing of
different regimes of interest. Here, we adopt the case of Tropical Western Pacific–
International Cloud Experiment (TWP-ICE), which was supported by the ARM
program of the Department of Energy and the Bureau of Meteorology of Australia from
January to February 2006 over Darwin in Northern Australia. Based on the
meteorological conditions, the TWP-ICE period can be divided into four shorter periods:
the active monsoon period (19–25 January), the suppressed monsoon period (26
January to 2 February), the monsoon clear-sky period (3–5 February), and the monsoon
break period (6–13 February, May *et al.* [2008]; Xie *et al.* [2010]). To take advantage
of previous studies of cloud-resolving models and single-column models, we followed
the setup of Franklin *et al.* [2012] to initiate the single-column runs starting on 19
January, 2006, and running for 25 days.
Stand-alone CAM5.3 simulations of the CESM model, forced by climatological sea
surface temperature for the year 2000 (*i.e.*, CESM compset: F_2000_CAM5), are



conducted to demonstrate global results. The horizontal resolution of the CESM global
runs is set at 2°. Individual global simulations are integrated for 12 years, and the output
for the last 10 years is used to calculate climatological means and annual cycles in
global means. Because we made changes largely with respect to CF, we also conducted
corresponding simulations using the satellite-simulator approach to provide CF for a
fair comparison with satellite CF products and typical CESM model output. This was
done using the CFMIP Observation Simulator Package (COSP) built into CESM 1.2.2
[Kay *et al.*, 2012]. In addition to the default monthly outputs, daily outputs of several
selected variables are also written out for more in-depth analysis.


3. Observational datasets and offline calculations

3.1 Observational data
Cloud field comparisons are critical for modifications to our system with respect to
cloud macrophysical schemes. Therefore, we use the products from
CloudSat/CALIPSO to provide CF and CWC data for evaluating the modeling
capabilities of the default and modified GTS cloud macrophysical schemes.
In addition to cloud observations, observational radiation fluxes from CERES-EBAF
are also used to investigate whether simulations using our system will improve radiation
calculations for both shortwave and longwave radiation flux, as well as their
corresponding cloud radiative forcings. Precipitation data are compared with Global
Precipitation Climatology Project data and several other climatic parameters, *e.g.*, air
temperature, RH, precipitable water, and zonal wind, are evaluated against the
reanalysis data (ERA-Interim).
We further evaluate the performance of the three macrophysics schemes by using the
approach of Su *et al.* [2013], which compares CF and CWC sorted by large-scale
dynamical and thermodynamic parameters. The CF products are based on the 2B-
GEOPROF R04 dataset [Marchand *et al.*, 2008], while the CWC data are based on the
2B-CWC-RO R04 dataset [Austin *et al.*, 2009]. The methodology from Li *et al.* [2012]
is used to generate gridded data. Four years of CloudSat/CALIPSO data, from 2007 to
2010, are used to carry out the statistical analyses. These data are used to obtain overall
climatological means to compare to those obtained from model simulations instead of
undergoing rigorous year-to-year comparisons between observations and simulations.
Monthly data from ERA-Interim for the same four years are used to obtain the
dynamical and thermodynamic parameters used in Su *et al.*'s approach. These
parameters include large-scale vertical velocity at 500 mb and RH at several vertical
levels.




3.2 Offline calculation of cloud fraction
To evaluate the impact of assumptions of CF distributions for the RH- and PDF-
based schemes, we conducted offline calculations of the CF by using the reanalyzed
temperature, humidity, and condensate data from ERA-Interim. As the differences in
CF characteristics do not change from month to month, the results for July are shown
in Figure 2 as an example. The ERA-Interim reanalysis performed by Dee *et al.* [2011]
using a 0.75° resolution from 1979 to 2012 is used in the calculation. With this offline
approach, we can observe the impacts of these macrophysics assumptions with a
balanced atmospheric state provided by the reanalysis.
Overall, the geographical distributions from the two GTS schemes are similar to that
of the ERA-Interim reanalysis shown in Figure 2. In July, high clouds corresponding to
deep convection are shown over South and East Asia where monsoons prevail. The
diagnosed clouds of the GTS scheme have a maximum level of 125 hPa, which is
consistent with those of the ERA-Interim reanalysis, but also have a more extensive
cloud coverage of up to 90%. Below the freezing level at approximately 500 hPa, the
CF diagnosed by the GTS scheme is comparable with that diagnosed by ERA-Interim
reanalysis. The most substantial differences in CF between the GTS scheme and the
ERA-Interim are observed in the mixed-phase clouds, such as the low clouds over the
Southern and Arctic Oceans. Such differences suggest that more complexity in
microphysics assumptions may be needed to describe the large-scale balance of mixed-
phase clouds. The diagnosed CF for the Park macrophysics scheme is also shown in the
right column of Figure 2. We found that the cloud field diagnosed by the Park
macrophysics scheme was considerably different from that diagnosed by ERA-Interim
reanalysis and the GTS schemes. The Park scheme diagnosed overcast high clouds of
100–125 hPa with coverage of up to 100% over the warm pool and Intertropical
Convergence Zone, but very little cloud coverage below 200 hPa, suggesting that the
assumptions of the Park scheme are probably not suitable for large-scale states of the
ERA-Interim reanalysis.
However, such a calculation does not account for the feedback of the clouds to the
atmospheric states through condensation or evaporation and cloud radiative heating.
Therefore, we further extended our single-column CAM5.3 experiments to examine the
impact of the cloud PDF assumption.


4. Single-column results
This section presents the analysis of single-column simulations using the TWP-ICE
field campaign. We focused on the CF fields and humidity fields, and their relation to





each other, to see how the $RH_c$ assumption affects these features through humidity partitioning. Three sets of model experiments were conducted. In addition to the T_pdf of the GTS and Park schemes, we also include the T_pdf of the GTS scheme with the Slingo ice CF parameterization. This experiment can help us to interpret the impacts of $RH_c$ on liquid and ice CFs separately.

Figure 3 shows the correlation between CF and RH for the three time periods during the TWP-ICE. As expected, the correlation coefficients are quite similar for the individual schemes during the active monsoon period when convective clouds dominated (R = 0.688, Park, vs. 0.698, T_pdf). In contrast, the correlation coefficient between CF and RH differs during the suppressed monsoon period when stratiform clouds dominated (R = 0.510, Park, vs. 0.728, T_pdf). The correlation coefficient between CF and RH is approximately 20% higher for the stratiform-cloud-dominated period when using T_pdf in the GTS scheme. It is also worth mentioning that, during the monsoon break period when both convective and stratiform clouds co-exist, the usage of the GTS scheme can also increase the correlation between CF and RH by 10% compared to the default Park scheme. The higher correlation coefficient for stratiform-cloud-dominated areas also suggests that the GTS scheme can somehow better simulate the variation of CF associated with RH, for which stratiform cloud macrophysics parameterization normally takes effect in CAM5.3.

Comparisons between T_pdf with the Slingo ice CF and the Park scheme can be used to examine the role of applying a PDF-based approach in simulating the liquid CF in the GTS scheme. The use of a PDF-based approach for calculating the liquid CF can increase the correlation between CF and RH by approximately 13% during the suppressed monsoon period (R = 0.637, T_pdf with Slingo, vs. 0.51, Park). Such an outcome also suggests that implementing a PDF-based approach for liquid clouds can lead to more reasonable fluctuations between CF and RH in GCM grids.

It turns out that using the PDF-based approach for ice clouds also contributes to the increased correlation between CF and RH by approximately 10%, as shown in Figure 3 with the T_pdf scheme (R = 0.637, T_pdf with Slingo, vs. 0.728, T_pdf). Such results also suggest that extending this PDF-based approach for ice clouds can better simulate changes in the cloud ice fraction using an RH-based approach rather than an $RH_c$-based approach. Such pair comparisons (*i.e.*, T_pdf with Slingo cloud ice fraction scheme vs. T_pdf and vs. Park) also reveal the important features of the GTS scheme, such as how variations in both the ice and liquid CF are better correlated with changes in RH of the GCM grids when compared to that of the default cloud macrophysics scheme.

Figure 4 shows scatter plots of RH and CF between 50 and 300 hPa determined from observations [Xie *et al.*, 2010] and simulated by models run for the suppressed monsoon period from the TWP-ICE case. It is evident that the relationship between CF and RH


appears to be more linear using the T_pdf of the GTS scheme as shown in Figure 4(c)
when compared to the default Park scheme (Figure 4(b)). Moreover, the CF-RH
distributions simulated by the GTS scheme are also closer to those of the observational
results except under more overcast conditions (*i.e.*, RH > 70% and RH > 110%). Similar
to the results shown in Figure 3, the role of a PDF-based treatment of CF for the liquid
CF can increase the degree of linearity between CF and RH (Figures 4(b) vs. 4(d)). On
the other hand, by excluding PDF-based treatment for the cloud ice fraction in the GTS
scheme, a more obvious spread in the CF-RH distribution is produced (comparing
Figures 4(c) and 4(d)). In other words, results from the paired comparisons shown in
Figure 4 are consistent with features shown in Figure 3, suggesting that applying a PDF-
based treatment for both liquid and ice CF parameterization can indeed increase the
linearity between CF and RH simulated by GCMs.


5. Global-domain results
5.1 Impacts on cloud fields
a. Cloud fraction

In Figure 5, total CF simulated by the GTS schemes and the CESM default cloud

macrophysics scheme, obtained from the COSP satellite simulator of the AMWG
package of NCAR CESM, are compared with the total CF in CALIPSO-GOCCP. Both
global mean and root-mean-square error (RMSE) values are improved by applying
U_pdf in the GTS scheme. The CF simulation resulting from the use of U_pdf in the
GTS scheme is qualitatively similar to that of CloudSat/CALIPSO, especially over the
mid- and high-latitude regions and for the annual and December-January-February
(DJF) simulations (Figure 6). On the other hand, the results of the Park scheme in the
tropics show clouds at higher altitudes than either U_pdf or T_pdf, in closer agreement
with CloudSat/CALIPSO. Cross-section comparison of the zonal height shows that the
CF simulation using U_pdf and T_pdf in the GTS scheme agrees better with that of
CloudSat/CALIPSO than that produced by Park under most scenarios (globally, within
60° N–60° S, and within 30° N–30° S), especially for the annual and DJF simulations
(Table 1). In contrast, some scenarios show lower RMSEs when the Park scheme is
used, *e.g.*, for the June-July-August (JJA) season globally, within 30–90° N, and within
30–90° S. Interestingly, when high latitudes are included (*i.e.*, 30–90° N and 30–90° S),
U_pdf still results in the smallest RMSE values, except for during the JJA season.

We also compared the annual latitude–longitude distributions of CF at different

specific pressure levels (Figure 7). The use of U_pdf resulted in a CF simulation
relatively similar to that of CloudSat/CALIPSO for mid-level clouds, *i.e.*, 300–700 mb,
particularly for the mid- and high latitudes. However, none of the CF parameterizations



are able to simulate stratocumulus clouds effectively, as revealed at the 850 and 900 mb
levels. For high clouds, the GTS and Park schemes exhibit observable differences
regarding the maximum CF level. Table 2 summarizes the RMSE values for the
latitude–longitude distribution of CFs at nine specific levels for the three schemes and
CloudSat/CALIPSO for the annual, JJA, and DJF means. For the annual mean, U_pdf
results in the smallest RMSE at all levels except at 125 mb, for which the Park scheme
yields the smallest RMSE (Table 2). For JJA, the Park scheme is closer to the
observations aloft (100–200 mb) and nearest the surface (900 mb). For DJF, U_pdf
again performs best at most levels except 100 and 125 mb, for which T_pdf is slightly
better, while for JJA, U_pdf is only best for most of the levels below 300 mb. Overall,
U_pdf in the GTS scheme results in better latitude–longitude CF distributions for 300–
900 mb for the annual, DJF, and JJA means, suggesting improvements in CF simulation
for middle and low clouds.
When annual, DJF, and JJA mean vertical CF profiles are averaged over the entire
globe and between 30° N and 30° S, U_pdf in the GTS scheme can produce a global
simulation close to that of CloudSat/CALIPSO for 200–850 mb (Figure S3). In contrast,
there is a large discrepancy between the simulated and observed CFs over the tropics.
Although the GTS schemes can simulate CF profiles above 100 mb, the height of the
maximum CF is lower than that of CloudSat/CALIPSO. In contrast, the height of the
maximum CF simulated by the Park scheme is similar to that of CloudSat/CALIPSO
but overestimated in CF. As before, when compared with CloudSat/CALIPSO, U_pdf
in the GTS scheme results in the smallest RMSE and the largest correlation coefficient
of the three schemes, whether or not the lower levels are included except in JJA at 125
mb, for which Park yields the smallest RMSE (Table 3). The reason for excluding the
lower levels from the statistical results is that there may be a bias for low clouds
retrieved by CloudSat due to radar-signal blocking by deep convective clouds.
The different degrees of changes for the global and tropical CFs can be attributed to
the relative roles of cumulus parameterizations (both deep and shallow) and stratus
cloud macrophysics/microphysics for the different latitudinal regions. It is expected that
the GTS scheme can alter CF simulations in the mid- and high-latitude areas more than
in the tropics because more stratiform clouds occur in those areas. It is also interesting
to note that, although it is known that more convective clouds exist in the tropics (*i.e.*,
the cumulus parameterization contributes more to the grid CF), the GTS scheme can
also affect the CF simulation over the tropics to some extent. Such different responses
in GCMs can be attributed to the degree of correlation between CF and RH for the
different types of clouds, as shown in the single-column model simulations (Figure 3).
This increase in the correlation coefficient between CF and RH is also evident in the
global simulations shown in Figure 8, where the two selected grids in the tropics are



examined. It is clear that the correlation coefficient ($R^2$) is approximately 20% higher
for high clouds when applying T_pdf or U_pdf in the GTS scheme rather than using
the default Park scheme ($R^2 = 0.410$, Park, vs. 0.652, T_pdf, vs. 0.646, U_pdf). An
increase in linearity between CF and RH with regard to high clouds can also be seen
for other latitudinal grids, as shown in Figures S1 and S2, especially for the high
latitudinal grids (Figure S3, $R^2 = 0.265$, Park, vs. 0.656, T_pdf, vs. 0.587, U_pdf). As
addressed in Section 4, this increase in the correlation coefficient between CF and RH,
also seen in the global simulation results, is contributed by the application of the PDF-
based approach for parameterizing both the liquid and ice CFs in the GTS scheme.

b. Cloud fraction and cloud water content
In Figures 9 and 10, the distributions of CWC and CF as functions of large-scale
vertical velocity at 500 mb (ω500) or mean RH averaged between 300 and 1000 mb
(RH300–1000) are evaluated against CloudSat/CALIPSO observations for 30° N–
30° S and 60° N–60° S. Figures 9 and 10 show that the model simulations are all
qualitatively more similar to each other than to the observations. Further statistical
comparisons are shown in Table 4. It is encouraging to note that, in addition to the slight
improvements in CF for both of these latitudinal ranges, the use of U_pdf in the GTS
scheme results in a CWC simulation that is more consistent with CloudSat/CALIPSO,
whether it is plotted against ω500 or RH300–1000. The RMSE and correlation
coefficient (R) values in Table 4 confirm this. For global simulations, using U_pdf also
results in better agreement with CloudSat/CALIPSO for both CF and CWC when they
are plotted against ω500, although for CWC plotted against RH300–1000, the Park
scheme yields the smallest RMSE (Table 4). Overall, these comparisons yield results
that are consistent with the general characteristics of most CMIP5 models, as found by
Su *et al.* [2013]. GCMs in general simulate the distribution of cloud fields better with
respect to a dynamical parameter as opposed to a thermodynamic parameter.
It is also worth noting that the use of U_pdf yields a 20–30% improvement in R when
plotted against RH300–1000 for the two latitudinal ranges, 30° N–30° S and 60° N–
60° S. The observable improvement in a thermodynamic parameter is an indication of
the uniqueness of this GTS scheme, in that it is capable of simulating the variation in
cloud fields relative to that in RH fields. There are also slight improvements in cloud
fields with respect to large-scale dynamical parameters. On the other hand, the Park
scheme results in an approximately 20% improvement in R when plotted against
RH300–1000 for the global domain, suggesting that the default Park scheme still
simulates cloud fields better over the high latitudinal regions. It is thus worth addressing
the likelihood that the different CF and CWC results for the different latitudinal ranges
simulated using the GTS scheme induce cloud–radiation interactions distinct from





those simulated in the Park scheme. Such changes in cloud–radiation interactions would
not only modify the thermodynamic fields but also the dynamic fields in the GCMs.
These changes are in turn likely to affect the climate mean state and variability. We
assess and compare these potential effects in the following subsection.
5.2 Effects on annual mean climatology
GTS schemes tend to produce smaller RMSE values for most of the global mean
values of the radiation flux, cloud radiative forcing, and CF parameters shown in Table
5, suggesting that the GTS scheme is capable of simulating the variability of these
variables. Furthermore, the assumed U_pdf shape appears to perform better for
outgoing longwave radiation flux, longwave cloud forcing (LWCF), and CF at various
levels, whereas the T_pdf assumption is better for simulating net and shortwave
radiation flux at the top of the atmosphere as well as shortwave cloud forcing (SWCF)
(Table 5). On the other hand, the Park scheme is better for simulating clear-sky net
shortwave radiation flux and precipitation. Smaller RMSE values can also be seen for
parameters such as total precipitable water, total-column cloud liquid water, zonal wind
at 200 mb (hereafter, U_200), and air temperature at 200 mb (hereafter, T_200) when
U_pdf of GTS is used. For global annual means, U_pdf simulates net radiation flux at
the top of the atmosphere, all- and clear-sky outgoing longwave radiation flux, and
precipitable water as well as U_200 and T_200 in closer agreement with observations.
In contrast, the Park scheme is better for simulating global mean variables such as net
shortwave radiation flux at the top of the atmosphere, longwave cloud forcing, and
precipitation. T_pdf simulates SWCF closest to the observational mean.
Overall, the averaged RMSE values of the ten parameters are 0.97 and 0.96 for U_pdf
and T_pdf, respectively, in the GTS schemes (Figure 11), suggesting that using the GTS
schemes would result in global simulation performances more or less similar to those
from the Park scheme. It is also worth noting that the biases in RH are smallest when
U_pdf in the GTS scheme is used (Table S1 of the supplementary material). In contrast,
T_pdf results in the smallest biases for SWCF, sea-level pressure, and ocean rainfall
within 30° N–30° S. On the other hand, the Park scheme produces the smallest biases
regarding mean fields such as LWCF, land rainfall within 30° N–30° S, Pacific surface
stress within 5° N–5° S, zonal wind at 300 mb, and temperature.
Comparisons of latitude–height cross-sections of RH and ERA-Interim show that the
GTS schemes tend to simulate RH values smaller than the default scheme does,
especially for high-latitude regions (> 60° N and 60° S), as shown in Figure 12. In
general, in terms of RH, using T_pdf in the GTS scheme results in better agreement
with ERA-Interim (Table 6). Figure 13 shows that the Park and T_pdf schemes are
wetter than ERA-Interim almost everywhere and that the uniform scheme is sometimes





drier. Table 7(a) further suggests that specific humidity simulated by the GTS schemes
is slightly more consistent with ERA-Interim than the Park scheme. Comparisons of air
temperature show that the three schemes tend to have cold biases almost everywhere.
However, it is interesting to note that the cold biases are reduced to some extent while
using the GTS schemes compared to the default scheme, as is evident in the smaller
values of RMSE shown in Table 7(b). These effects on moisture and temperature are
likely to result in changes in the annual cycle and seasonality of climatic parameters.
Such observable changes in RH, clouds (both CF and CWC), and cloud forcing suggest
that the GTS scheme will simulate cloud macrophysics processes in GCMs quite
differently from the Park scheme, owing to the use of a variable-width PDF that is
determined based on grid-mean information.
5.3 Changes in the annual cycle of climatic variables
Figure 14 shows the annual cycle of precipitable water simulated by the three
schemes. The magnitude of precipitable water simulated by the GTS schemes is closer
to the ERA-Interim data than the Park simulation is (Table 8). Interestingly, U_pdf
results in slightly better agreement with ERA-Interim than T_pdf for the region 60° N–
60° S. This implies that the GTS scheme would alter the moisture field for both RH and
precipitable water in GCMs. These results are relatively more realistic with respect to
both the moisture field and CF and CWC (Figures 9 and 10) and are likely to yield a
more reasonable cloud–radiation interaction in the GCMs. It is therefore also worth
examining any differences in dynamic fields, for example, in the annual U_200 cycle,
between the three schemes and the ERA-Interim data (Figure 15). Like the annual cycle
of precipitable water, U_200 simulated by the GTS schemes is closer to that of ERA-
Interim than that simulated by the Park scheme, as supported by the smaller RMSE
shown in Table 8. Furthermore, the U_pdf assumption results in a better annual U_200
cycle than the T_pdf assumption, especially for 60° N–60° S. This further supports the
argument that this GTS scheme can effectively modulate global simulations, with
respect to both thermodynamic and dynamical climatic variables.
Figure 16 displays the global mean annual cycles of several parameters simulated by
the three schemes and the corresponding parameters from observational data. The GTS
scheme simulations of total precipitable water (TMQ) are close to that of ERA-Interim;
indeed, U_pdf almost exactly reproduces the ERA-Interim TMQ. However, we must
admit that such good agreement of the global mean is partly due to offsetting wet and
dry differences from ERA-Interim. The GTS schemes also produce a more reasonable
global mean annual cycle for outgoing longwave radiation (FLUT). It is probably due
to the reduced CF simulated by the GTS scheme compared to the Park scheme even
though the cloud top heights simulated by GTS are lower than observations in the





tropics. Interestingly, for SWCF, T_pdf yields a simulation closer to the observations than the other two schemes, which is consistent with the features of the global annual mean of SWCF shown in Figure 11 and Table S1. However, for LWCF, the annual cycle simulated by Park is closest to the observations. The U_pdf of the GTS scheme also results in improvements in U_200 and T_200 (Figure 16). The RMSEs for all of these comparisons confirm these results (Table 9).

5.4 Changes in cloud–radiation interactions

As mentioned in Section 5.1, usage of the GTS cloud macrophysics schemes would affect the cloud fields, *i.e.*, CF and CWC. This, in turn, is likely to affect global simulations with respect to both mean climatology and the annual cycles of many climatic parameters (as discussed in Sections 5.2 and 5.3) through cloud–radiation interactions. Figure 17 compares CF, radiation heating rate (*i.e.*, longwave heating rate plus shortwave heating rate, hereafter QRL+QRS) and temperature trends due to moist processes (hereafter, DTCOND) for each pair-wise combination of the three schemes. Qualitatively consistent changes in CF are apparent for the GTS schemes, *e.g.*, an increase in the highest clouds over the tropics and a decrease below them, a decrease in 150–400 mb clouds over the mid-latitudes, a decrease in 300–700 mb clouds over the high latitudes, an increase in 300–700 mb clouds over the tropics to mid-latitudes, and an increase in low clouds over the high-latitude regions. The GTS schemes also yield a significant increase in CF at atmospheric levels higher than 300 mb over the high-latitude regions (Figure 17). These changes affect the radiation calculations to some extent. In addition, CWC is also affected by the GTS schemes (Figures 9 and 10). The combined effects of the changes in CF and CWC are likely to result in changes in cloud–radiation interactions. In addition, although there are significant changes in CF at high atmospheric levels in the high-latitude regions, the combined effect of CF and CWC on QRL+QRS is quite small, owing to the low CWC values over this region. The changes in moisture processes, *i.e.*, DTCOND (Figure 17), also suggest that the combined effects of the changes in the thermodynamic and dynamical fields occur as a result of changes in cloud–radiation interactions within the GCMs from GTS schemes.

The bottom panel in Figure 17 shows the differences in CF, QRL+QRS, and DTCOND between the two GTS schemes. Relative to T_pdf, U_pdf simulates a greater CF for 300–1000 mb clouds within 60° N–60° S, but a smaller CF for all three cloud levels for the high-latitude regions. Furthermore, the CWC vertical cross-section also differs for the two GTS schemes (data not shown for limitations of space). Combining the changes in CF and CWC, the corresponding changes in QRL+QRS and DTCOND, particularly the increase of low clouds over the mid-latitude region, are clear with an obvious decrease of high clouds over the tropical to mid-latitude region.



Observable changes in large-scale circulations are likely, given the various changes
in QRL+QRS and DTCOND resulting from applying different cloud macrophysics.
Accordingly, both the mean and variability of the climate simulated by the GCMs differ
among the three schemes, as shown in the previous subsections. These results
emphasize the importance of improving cloud-related parameterization to provide
better simulations of the cloud–radiation interaction within GCMs. Furthermore, as
previously shown, the cloud–radiation interaction is highly sensitive to the assumptions
of the CF parameterization used in the macrophysical scheme in the GCMs, even if
there is only a small change in the CF parameterization. The uniqueness of the GTS
scheme is in its application of a variable PDF width to calculate CF in the default PDF-
based CF scheme of the CESM model. Further systematic experiments are necessary to
improve our understanding of the sensitivity of the GTS scheme, and some are
presented in Section 5.6.

5.5 Consistent changes in cloud radiative forcing, cloud fraction, and cloud condensates
Observable changes in clouds and radiation fluxes after adopting the GTS scheme
were clearly shown in the previous subsections. It is thus worth examining features in
cloud radiative forcings caused by the GTS scheme that produce such changes, as
compared to those of the default Park scheme. Figure 18 shows the difference in total
cloud fraction, SWCF, LWCF, CF, and averaged cloud water contents, as well as the
averaged RH at the three levels i.e., 100–400, 400–700, and 700–1000 mb, derived
from the T_pdf of GTS with the Park results subtracted. One can readily observe that
changes in SWCF (Figure 18(b)) are quite consistent with those for total CF, showing
a decrease in the total CF over the area within 30° N and 30° S with an increase
everywhere else (Figure 18(a)). Such prominent changes in latitudinal distribution of
SWCF can be further related to the changes in the low (Figure 18(e)) and middle (Figure
18(f)) CFs particularly associated with low clouds.
On the other hand, changes in the high CF (Figure 18(d)) are also quite consistent
with those in LWCF (Figure 18(c)), showing an overall decrease of high clouds
especially over the tropical convection areas. As expected, changes in cloud water
condensates (Figures 18(g)–(i)) are closely related to changes in the CF at the three
levels except for the middle clouds. Therefore, according to the evidence shown in
Figures 18(a)–(i), it is clear that use of the GTS scheme would cause significant changes
in the spatial distribution of low, middle, and high clouds (both in CF and cloud water
condensates) that would result in corresponding changes in cloud radiative forcings
(both for SWCF and LWCF).
Surprisingly, changes in RH at the three levels (Figures 18(j)–(l)) are relatively less
consistent with changes in the CF and condensates, especially for middle and low





clouds over the mid- and high-latitude areas. Such results also indicate that there are
complicated factors accounting for changes in RH in the GCMs. We suggest that, in
addition to the active roles of the GTS scheme in redistributing/modulating moisture
between clouds (*i.e.*, cloud liquid or ice) and environment (water vapor) in GCM grids,
thermodynamic and dynamical feedback resulting from cloud–radiation interactions
also contribute to RH changes. At the present stage, we cannot quantify these individual
contributions. More in-depth analysis is needed to unveil the detailed mechanisms of
why GTS schemes tend to produce less low clouds over the tropics while more low
clouds over the mid- and high latitudes compared to the default Park scheme, as well
as observable changes regarding middle and high clouds.
5.6 Uncertainty in GTS cloud fraction parameterization
a. Assumption of PDF shape in the GTS scheme
In general, the simulations of CF, RH, and other parameters (*e.g.*, global annual mean
and/or annual cycle) using the T_pdf scheme that have been discussed and illustrated
thus far have distribution features qualitatively and values quantitatively between those
of the Park and U_pdf schemes. In other words, the characteristics of the T_pdf
simulations are a combination of those from both the default Park scheme and the
U_pdf scheme. This is to be expected because there are fewer differences between the
Park and T_pdf schemes than between the Park and U_pdf schemes in terms of cloud
macrophysics parameterization. Since the shape of the PDF is triangular for both the
Park and T_pdf schemes, the only difference between these two is that T_pdf has a
variable PDF width that is based on the grid-mean mixing ratio of hydrometeors and
the saturation ratio of the atmospheric environment, rather than the fixed-width function
of $RH_c$. Even such a minor difference, however, can have an impact on both the
thermodynamic and dynamical fields in global simulations. Our findings further
suggest that the use of a variable PDF width to determine CF results in some changes
in consistency between the RH and CF fields, as well as in the simulation of SWCF and
net radiation flux at the top of atmosphere. As mentioned in Section 1, a diagnostic
approach to determining the triangular PDF width of the default Park scheme can be
used to refine the Park scheme [Appendix A of Park *et al*., 2014]. This is effectively the
same as using the GTS scheme with T_pdf.
However, it is also evident that assuming a uniform PDF (*i.e.*, a rectangular shape)
can have a larger effect on global simulations, as seen with our use of U_pdf. It is
interesting to note that the use of U_pdf yields a smaller overall RMSE for many
thermodynamic and dynamical fields than does the use of T_pdf. This implies that a
uniform distribution is probably more appropriate for the 2° horizontal resolution
currently used in global simulations. The scale-dependence of the PDF shape is





certainly important to consider, as revealed in our comparisons between T_pdf and
U_pdf, but this is beyond the scope of this paper. Furthermore, the possible dependence
of PDF shape on specific cloud systems in different regions should also be examined
using systematic tests and simulation designs.
b. Uncertainty resulting from cloud-ice fraction parameterization
It is worth evaluating the possible uncertainty related to CF for cloud ice because the
saturation adjustment assumption used for cloud liquid may not apply to cloud ice, as
discussed in Section 1. We thus examine the sensitivity of the super-saturation values
for the cloud ice fraction by multiplying by $q_{si}$, as shown in equation 4 by the constant
*sup*. Several values of *sup* are assumed for the cloud ice fraction in the GTS schemes
with CF simulated using Slingo's approach to parameterization as used by Park *et al*.
[2014] and are compared with the CloudSat/CALIPSO observational data (Figure S4).
Both GTS schemes are sensitive to the *sup* value. For U_pdf, CF decreases more-or-
less linearly with increasing *sup* values, but there is no such clear linearity for T_pdf,
especially for *sup* values of 1.0000–1.0005. Interestingly, changing the *sup* value for
the cloud ice fraction affects the cloud liquid fraction results for the scheme. We also
find that the CF profile simulated by U_pdf when *sup* = 1.0005 is similar to that
simulated using Slingo's approach to parameterization, especially for middle and low
clouds. Based on these sensitivity tests, it is evident that the *sup* value used in the cloud
ice fraction formulae of the GTS scheme can be regarded as a tunable parameter under
the present cloud macrophysics and microphysics framework of the CESM model.
When *sup* = 1.0 in the GTS scheme with U_pdf, the results are comparable to
CloudSat/CALIPSO observations, while with T_pdf, the *sup* value can be tuned
between 1.0 and 1.005 to mimic the CloudSat/CALIPSO data (Figure S4). Thus, the
results of GTS schemes are sensitive to the supersaturation threshold and suggest that
it is still quite challenging to produce a reasonable parameterization for the cloud ice
fraction, given the longer time-scales needed for ice clouds to reach saturation
equilibrium.
6. Conclusions
In this paper, we presented a macrophysics parameterization based on a probability
density function (PDF) called the GFS-TaiESM-Sundqvist (GTS) cloud macrophysics
scheme, which is based on Sundqvist's cloud macrophysics concept for global models
and the recent modification of the cloud macrophysics in the NCAR CESM model by
Park *et al*. [2014]. The GTS scheme especially excludes the assumption of a prescribed
critical relative humidity threshold ($RH_c$), which is included in the default cloud
macrophysics schemes, by determining the width of the PDF based on grid





hydrometeors and saturation ratio.
We first used ERA-Interim reanalysis data to examine offline the validity of the
relationship between cloud fraction (CF) and relative humidity (RH) based on the PDF
assumption. Results showed that the GTS assumption better describes the large-scale
equilibrium between CF and environment conditions. In a single-column model setup,
we noticed several improved characteristics of the CF and humidity in the model
simulation when the *ad hoc* $RH_c$ assumptions were removed. First, the CF-RH
relationship is more consistent in our modified scheme and an overall increase in
linearity between CF and RH of approximately 20% was observed, especially for
stratiform clouds. Second, according to the pair-wise comparisons shown and discussed
in Figures 3 and 4, the use of PDF-based treatments for parameterizing both liquid and
ice CFs in the GTS schemes contributed to the linear CF-RH relationship.
According to our detailed comparisons with observational cloud field data (CF and
cloud water content (CWC)) from CloudSat/CALIPSO, GTS parameterization is able
to simulate changes in CF that are associated with changes in RH in global simulations.
Improvements with respect to the CF of middle clouds, the boreal winter, and mid- and
high latitudes are particularly evident. Furthermore, examination of the vertical
distributions of CF and CWC as a function of large-scale dynamical and
thermodynamic parameters suggests that, compared to the default scheme, simulations
of CF and CWC from the GTS scheme are qualitatively more consistent with the
CloudSat/CALIPSO data. It is particularly encouraging to observe that the GTS scheme
is also capable of substantially increasing the pattern correlation coefficient of CF and
CWC as a function of a large-scale thermodynamic parameter (*i.e.*, RH300–1000).
These effects appear to have a substantial impact on global climate simulations via
cloud–radiation interactions.
The fact that CF and CWC simulated by the GTS scheme are temporally and spatially
closer to those of the observational data suggests that not only the climatological mean
but also the annual cycles of many parameters would be better simulated by the GTS
cloud macrophysical scheme. Improvements with respect to thermodynamic fields such
as upper-troposphere and lower-stratosphere temperature, RH, and total precipitable
water were more substantial even than those in the dynamical fields. This was
consistent with our comparisons based on the vertical distribution of CF and CWC as
functions of large-scale dynamical and thermodynamic forcing. Interestingly, the GTS
scheme results in observable changes in the annual cycle of zonal wind at 200 hPa,
which suggests that the modification of thermodynamic fields resulting from changes
in cloud–radiation interactions will, in turn, reciprocally affect the dynamical fields.
Accordingly, it is worth investigating possible changes in large-scale circulation,
monsoon evolution, and short- and long-term climate variability in future research.
GTS schemes can simulate spatial distributions of cloud radiative forcings (both for
shortwave and longwave) quite differently compared to the default Park scheme.
Changes in cloud radiative forcings are very consistent with different latitudinal
changes in CF and cloud water condensates at the three cloud levels. The most
important feature of the GTS scheme is that CF is self-consistently determined based
on hydrometeors and the environmental information in the model grid box in the
general circulation model (GCM) simulation. In contrast to the prescribed vertical
profile of $RH_c$ used in many current GCMs, the width of the PDF in the GTS scheme is
variable and calculated in a diagnostic way. A fixed $RH_c$ is thus no longer used once
clouds are formed. This feature also potentially makes the GTS scheme a candidate
macrophysics parameterization for use in modern global weather forecasting and
climate prediction models as it better simulates the CF-RH relationship. However,
further efforts are required to develop a more meaningful and physical way to
parameterize the super-saturation ratio assumption applied to the cloud ice fraction in
the GTS scheme, and to investigate why a uniform PDF in the GTS scheme performs
better overall than the triangular PDF.

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





Code availability
The codes of the GTS scheme used in this study can be obtained from the following
website:
https://doi.org/10.5281/zenodo.3626654
Author contributions. HHH is the initiator and primary investigator of the TaiESM
project. CJS developed code and wrote the majority of the paper. YCW also developed
code and wrote part of the paper. WTC helped process CloudSat/CALIPSO satellite
data. HLP and RS helped develop the theoretical basis of the GTS scheme. YHC helped
with the off-line calculations. CAC helped with most of the visualizations.
Competing interests. The authors declare that they have no conflict of interest.






Acknowledgements
We would like to dedicate this paper to Dr. Chia Chou in appreciation for his
encouragement. The work is supported in part by the Ministry of Science and
Technology (MOST), Taiwan (R.O.C.) under the projects MOST 100-2119-M-001-
029-MY5 and 105-2119-M-002-028-MY3. This work is also part of the Consortium
for Climate Change Study (CCliCs) – Laboratory for Climate Change Research.
CloudSat data is available through Austin *et al.* [2009]. Other observations, satellite
retrievals, and reanalysis data used in the paper were obtained from the AMWG
diagnostic package provided by CESM, NCAR. Detailed information regarding those
observational data are available at
http://www.cgd.ucar.edu/amp/amwg/diagnostics/plotType.html. We would like to
thank Anthony Abram (www.uni-edit.net) for editing and proofreading this manuscript.







**Table 1.** Root-mean-square errors (RMSE) for comparisons of latitude–height cross-sections of CF among the three macrophysical schemes (Park: default scheme; T_pdf: triangular PDF in the GTS scheme; U_pdf: uniform PDF in the GTS scheme) and observational data from CloudSat/CALIPSO (Figure 6). Comparisons are made of the means for five latitudinal ranges and three periods (JJA: June, July, August; DJF: December, January, February). The smallest RMSE value of the three schemes in each case is bold and underlined.

| | Global | | | 60°N~60°S | | | 30°N~30°S | | | 30°N~90°N | | | 30°S~90°S | | |
|---|---|---|---|---|---|---|---|---|---|---|---|---|---|---|---|
| | Park | T_pdf | U_pdf | Park | T_pdf | U_pdf | Park | T_pdf | U_pdf | Park | T_pdf | U_pdf | Park | T_pdf | U_pdf |
| **Annual** | 7.15 | 8.27 | **6.75** | 5.25 | **4.53** | 4.85 | 5.84 | 5.37 | **5.05** | 8.78 | 10.40 | **8.52** | 6.46 | 8.29 | **6.18** |
| **JJA** | **7.40** | 11.30 | 9.50 | 6.27 | 5.64 | **5.61** | 6.03 | 5.96 | **5.56** | **8.91** | 10.60 | 9.13 | **6.93** | 15.50 | 12.70 |
| **DJF** | 9.04 | 9.37 | **6.99** | 5.62 | **5.24** | 5.38 | 6.29 | 5.53 | **5.36** | 12.80 | 13.00 | **10.00** | 6.33 | 7.85 | **3.82** |

**Table 2.** RMSEs for comparisons between CF at nine pressure levels, as simulated by the three macrophysical schemes (Park, T_pdf, U_pdf) and observational data from CloudSat/CALIPSO (Figure 7). The comparisons are made for three periods (JJA: June, July, August; DJF: December, January, February). The smallest RMSE value of the three schemes in each case is bold and underlined.

| | **Annual** | | | **JJA** | | | **DJF** | | |
|---|---|---|---|---|---|---|---|---|---|
| | **Park** | **T_pdf** | **U_pdf** | **Park** | **T_pdf** | **U_pdf** | **Park** | **T_pdf** | **U_pdf** |
| **100 mb** | 6.07 | 5.40 | **4.71** | **4.85** | 12.70 | 10.10 | 7.88 | **3.94** | 4.20 |
| **125 mb** | **4.70** | 5.56 | 4.80 | **6.13** | 12.60 | 10.10 | 5.96 | **4.56** | 4.81 |
| **200 mb** | 7.23 | 8.34 | **6.78** | **9.80** | 14.90 | 11.90 | 8.64 | 6.57 | **6.46** |
| **300 mb** | 10.80 | 9.63 | **7.98** | 11.60 | 12.90 | **10.80** | 12.40 | 11.70 | **9.06** |
| **400 mb** | 11.80 | 10.50 | **6.93** | 12.40 | 10.50 | **9.55** | 12.70 | 13.90 | **8.06** |
| **500 mb** | 11.00 | 11.50 | **7.65** | 11.90 | 10.60 | **9.28** | 11.70 | 13.40 | **8.50** |
| **700 mb** | 8.64 | 9.47 | **8.19** | 9.63 | 10.80 | **9.46** | 10.70 | 11.10 | **9.41** |
| **850 mb** | 14.30 | 14.20 | **12.00** | 14.80 | 15.40 | **12.80** | 16.10 | 15.30 | **13.20** |
| **900 mb** | 12.50 | 15.10 | **12.30** | **13.30** | 16.60 | 13.60 | 15.10 | 16.40 | **12.90** |





**Table 3.** RMSEs and correlation coefficients (R, in brackets) for comparisons between the vertical CF profiles simulated by the three macrophysical schemes (Park, T_pdf, U_pdf) and observational data from CloudSat/CALIPSO (Figure S3). Comparisons are made for three periods (JJA: June, July, August; DJF: December, January, February) and two latitudinal ranges. The smallest RMSE value of the three schemes in each case is bolded and underlined.

| | | **Global** | | | **30°N~30°S** | | |
|---|---|---|---|---|---|---|---|
| | | **Park** | **T_pdf** | **U_pdf** | **Park** | **T_pdf** | **U_pdf** |
| **Including low levels** | Annual | 8.03 (0.83) | 9.51 (0.86) | **7.92** (0.87) | 6.15 (0.60) | 6.03 (0.53) | **5.38** (0.59) |
| | JJA | **8.58** (0.81) | 11.52 (0.85) | 9.66 (0.85) | 6.61 (0.61) | 6.06 (0.54) | **5.87** (0.62) |
| | DJF | 9.14 (0.81) | 10.20 (0.85) | **7.92** (0.86) | 6.31 (0.59) | 6.65 (0.52) | **5.60** (0.57) |
| **Excluding low levels** | Annual | 5.97 (0.91) | 6.51 (0.96) | **5.32** (0.99) | 5.89 (0.63) | 5.75 (0.51) | **5.13** (0.55) |
| | JJA | **6.60** (0.92) | 9.39 (0.97) | 7.72 (0.98) | 6.13 (0.63) | 6.22 (0.49) | **5.55** (0.58) |
| | DJF | 6.05 (0.92) | 6.76 (0.95) | **4.85** (0.99) | 6.20 (0.60) | 5.93 (0.49) | **5.45** (0.52) |

**Table 4.** (a) RMSE and (b) R values for comparisons between CF and CWC simulated by the three macrophysical schemes (Park, T_pdf, and U_pdf) and plotted against vertical velocity at 500 mb ($\omega 500$) or averaged RH for 300–1000 mb (RH300–1000, obtained from the ERA-Interim reanalysis) and observational data from CloudSat/CALIPSO (Figures 9 and 10). The comparisons are made for three latitudinal ranges. The smallest RMSE or largest R value of the three schemes in each case is bolded and underlined.

(a)

| **RMSE** | | **Global** | | | **60°N~60°S** | | | **30°N~30°S** | | |
|---|---|---|---|---|---|---|---|---|---|---|
| | | **Park** | **T_pdf** | **U_pdf** | **Park** | **T_pdf** | **U_pdf** | **Park** | **T_pdf** | **U_pdf** |
| **OMEGA@500 mb** | CWC | 11.10 | 10.90 | **9.83** | 11.40 | 11.20 | **10.10** | 14.10 | 13.80 | **12.50** |
| | CF | 7.65 | 7.26 | **6.13** | 7.55 | 7.23 | **6.24** | 8.13 | 8.07 | **7.21** |
| **RH@300-1000 mb** | CWC | **8.73** | 9.69 | 11.60 | 13.50 | 15.10 | **11.80** | 19.10 | 18.00 | **12.00** |
| | CF | 17.90 | 18.30 | **13.90** | 15.40 | 17.30 | **12.70** | 18.80 | 18.30 | **12.90** |

(b)

| **R** | | **Global** | | | **60°N~60°S** | | | **30°N~30°S** | | |
|---|---|---|---|---|---|---|---|---|---|---|
| | | **Park** | **T_pdf** | **U_pdf** | **Park** | **T_pdf** | **U_pdf** | **Park** | **T_pdf** | **U_pdf** |
| **OMEGA@500 mb** | CWC | 0.73 | 0.77 | **0.80** | 0.74 | 0.77 | **0.80** | 0.60 | 0.66 | **0.74** |
| | CF | 0.84 | 0.85 | **0.89** | 0.85 | 0.85 | **0.88** | 0.83 | 0.82 | **0.84** |
| **RH@300-1000 mb** | CWC | **0.64** | 0.54 | 0.45 | 0.44 | 0.34 | **0.62** | 0.22 | 0.25 | **0.55** |
| | CF | 0.31 | 0.40 | **0.59** | 0.51 | 0.46 | **0.68** | 0.45 | 0.45 | **0.66** |





**Table 5.** Global annual means (Mean) and RMSE values for comparisons with the observed values (Obs) for a selection of climatic parameters simulated by the three cloud macrophysical schemes (Park, T_pdf, and U_pdf). The smallest RMSE value or closest global mean of the three schemes in each case is bolded and underlined.

| Parameters | Obs | Mean (Park) | Mean (T_pdf) | Mean (U_pdf) | RMSE (Park) | RMSE (T_pdf) | RMSE (U_pdf) |
|---|---|---|---|---|---|---|---|
| RESTOA_CERES-EBAF | 0.81 | 4.18 | 3.25 | **-1.06** | 12.39 | **10.43** | 11.11 |
| FLUT_CERES-EBAF | 239.67 | 234.97 | 237.88 | **238.14** | 8.78 | 6.73 | **6.50** |
| FLUTC_CERES-EBAF | 265.73 | 259.06 | 259.65 | **260.45** | 7.55 | 7.12 | **6.48** |
| FSNTOA_CERES-EBAF | 240.48 | **239.15** | 241.14 | 237.08 | 13.97 | **11.64** | 12.79 |
| FSNTOAC_CERES-EBAF | 287.62 | **291.26** | 291.31 | 291.70 | **7.08** | 7.09 | 7.58 |
| LWCF_CERES-EBAF | 26.06 | **24.10** | 21.77 | 22.31 | 6.78 | 6.77 | **6.21** |
| SWCF_CERES-EBAF | -47.15 | -52.11 | **-50.18** | -54.61 | 15.98 | **12.90** | 15.43 |
| PRECT_GPCP | 2.67 | **2.97** | 3.04 | 3.14 | **1.09** | 1.10 | 1.15 |
| PREH2O_ERAI | 24.25 | 25.64 | 24.90 | **24.45** | 2.56 | 2.05 | **2.03** |
| CLDTOT_Cloudsat+CALIPSO | 66.82 | **64.11** | 70.77 | 70.09 | 9.87 | 11.38 | **9.76** |
| CLDHGH_Cloudsat+CALIPSO | 40.33 | 38.17 | 44.79 | **40.22** | 9.37 | 9.28 | **8.17** |
| CLDMED_Cloudsat+CALIPSO | 32.16 | 27.22 | 30.41 | **31.26** | 8.03 | 6.95 | **6.28** |
| CLDLOW_Cloudsat+CALIPSO | 43.01 | **43.63** | 43.67 | 46.19 | **12.78** | 18.06 | 16.17 |
| CLDTOT_CALIPSO GOCCP | 67.25 | 56.43 | 55.45 | **61.72** | 14.38 | 15.37 | **10.28** |
| CLDHGH_CALIPSO GOCCP | 32.04 | **25.57** | 22.48 | 24.46 | **9.04** | 11.30 | 10.16 |
| CLDMED_CALIPSO GOCCP | 18.09 | 11.21 | 14.55 | **18.19** | 8.35 | 6.34 | **6.02** |
| CLDLOW_CALIPSO GOCCP | 37.95 | 33.24 | 33.16 | **38.41** | 10.63 | 11.33 | **9.98** |
| TGCLDLWP(ocean) | 79.87 | 42.55 | 40.68 | **48.74** | 40.92 | 42.37 | **35.16** |
| U_200_MERRA | 15.45 | 16.18 | 15.87 | **15.66** | 2.52 | 2.11 | **1.94** |
| T_200_ERAI | 218.82 | 215.58 | 215.76 | **216.84** | 4.03 | 3.37 | **2.13** |

**Table 6.** RMSEs for comparisons between the latitude–height cross-sections of RH simulated by the three macrophysical schemes (Park, T_pdf, and U_pdf) and ERA-Interim (Figure 12). The comparisons are made for three periods (JJA: June, July, August; DJF: December, January, February) and two latitudinal ranges. The smallest RMSE value of the three schemes in each case is bolded and underlined.

| RH | Park | T_pdf | U_pdf |
|---|---|---|---|
| Annual | 11.2 | **6.4** | 9.4 |
| JJA | 11.2 | **7.3** | 10.1 |
| DJF | 11.8 | **6.9** | 9.7 |





**Table 7.** RMSEs for comparisons between the latitude–height cross-sections of (a) specific humidity q and (b) air temperature T simulated by the three macrophysical schemes (Park, T_pdf, and U_pdf) and ERA-Interim (Figure 13). The comparisons are made for three periods (JJA: June, July, August; DJF: December, January, February) and two latitudinal ranges. The smallest RMSE value of the three schemes in each case is bolded and underlined.

(a)

| q | Park | T_pdf | U_pdf |
|---|---|---|---|
| Annual | 0.29 | 0.25 | **0.23** |
| JJA | 0.32 | **0.26** | 0.27 |
| DJF | 0.29 | 0.27 | **0.25** |

(b)

| T | Park | T_pdf | U_pdf |
|---|---|---|---|
| Annual | 2.62 | 2.49 | **2.05** |
| JJA | 2.65 | 2.43 | **2.24** |
| DJF | 2.94 | 2.86 | **2.60** |

**Table 8.** RMSEs for comparisons between the annual cycles of zonal mean total precipitable water (TMQ) and annual cycles of zonal wind at 200 mb (U200) simulated by the three macrophysical schemes (Park, T_pdf, and U_pdf) and ERA-Interim (Figures 14 and 15).

| | Park | T_pdf | U_pdf |
|---|---|---|---|
| TMQ | 1.44 | 0.86 | **0.82** |
| U200 | 1.97 | 1.74 | **1.49** |





**Table 9.** RMSEs for comparisons between the global mean annual cycles of several parameters simulated by the three macrophysical schemes (Park, T_pdf, and U_pdf) and corresponding observational data (Figure 16). The smallest RMSE value of the three schemes in each case is bolded and underlined.

|       | Park  | T_pdf     | U_pdf    |
|-------|-------|-----------|----------|
| TMQ   | 2.08  | **_1.70_**| 1.74     |
| FLUT  | 8.15  | 6.71      | **_6.31_** |
| LWCF  | 6.18  | 6.32      | **_6.06_** |
| SWCF  | 14.00 | **_11.80_** | 14.00  |
| U_200 | 2.34  | 2.04      | **_1.70_** |
| T_200 | 5.57  | 4.50      | **_2.55_** |





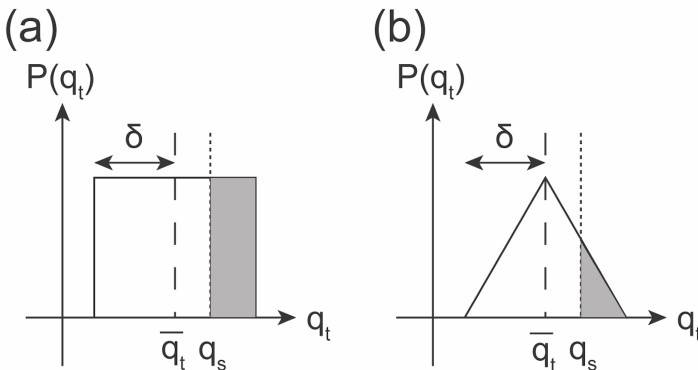

**Figure 1.** Illustration of sub-grid PDF of total water substance $q_t$ with (a) uniform distribution and (b) triangular distribution. The shaded part shows the saturated cloud fraction, $\delta$ represents the width of the PDF, $\overline{q_t}$ denotes the grid-mean value of total water substance, and $q_s$ represents the saturation mixing ratio as the temperature is assumed to be uniform within the grid.

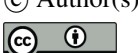

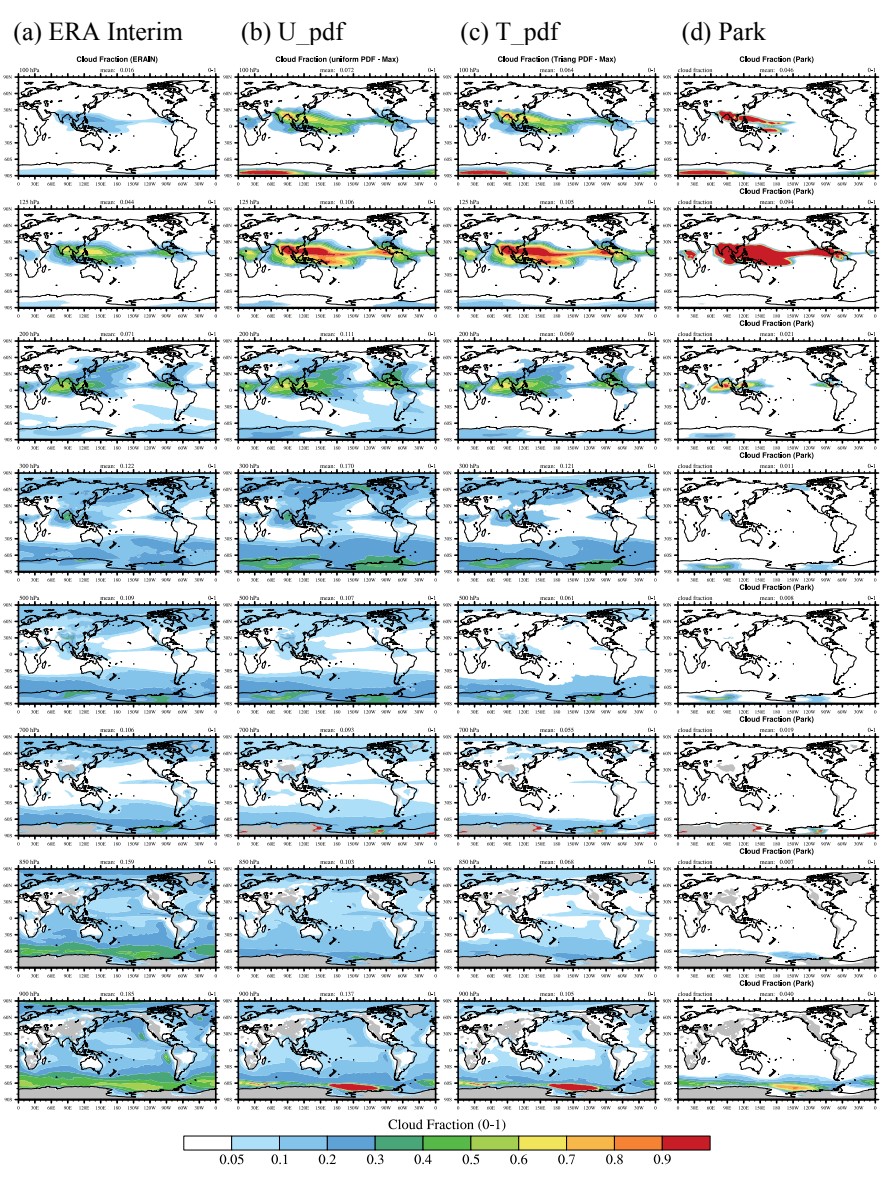

**Figure 2.** Mean cloud fraction in July (a) from the ERA-Interim reanalysis dataset and (b, c, d) diagnosed from cloud fraction schemes, with temperature, moisture, and condensates from the ERA-Interim reanalysis provided. From left to right, these schemes are the (b) U_pdf, (c) T_pdf, and (d) Park macrophysics schemes. Cloud distributions from 100 to 900 hPa are plotted from top to bottom.



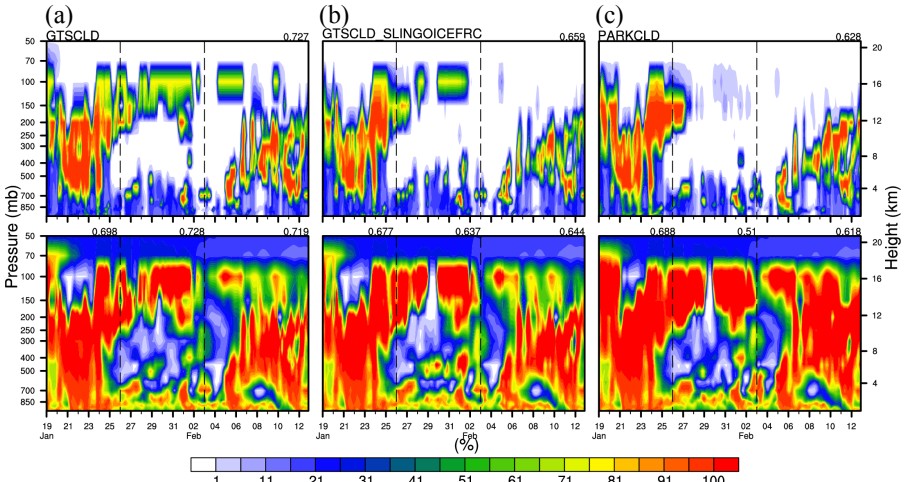

**Figure 3.** Pressure–time cross-sections of cloud fraction (upper row) and relative humidity (lower row) simulated by the three different cloud macrophysical schemes; (a) T_pdf, (b) T_pdf with Slingo ice cloud fraction scheme, and (c) the Park scheme during the TWP-ICE field campaign.





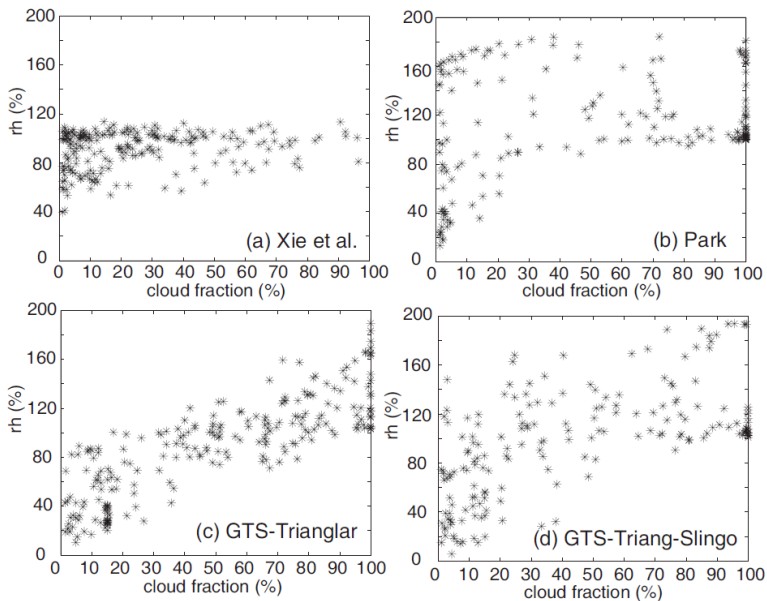

**Figure 4.** Scatter plots of high-level (50–300 hPa) relative humidities and cloud fractions during the suppressed monsoon period of the TWP-ICE field campaign (26 January to 2 February, 2006) observed by (a) Xie *et al*. [2010] and simulated by SCAM with the (b) Park of CAM5.3, (c) T_pdf, and (d) T_pdf with Slingo ice cloud macrophysics schemes.

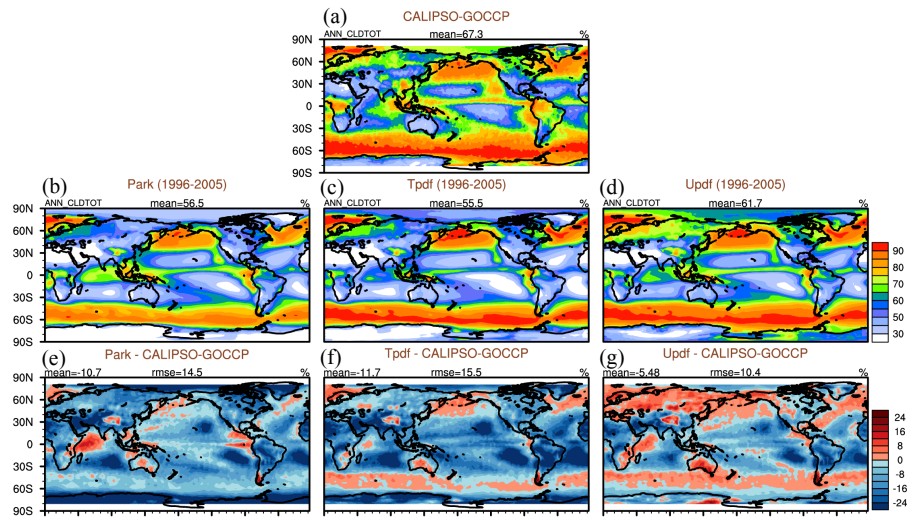

**Figure 5.** Total cloud fraction (CF) from (a) CALIPSO-GOCCP and simulated by the three schemes: (b) the default Park, (c) T_pdf, and (d) U_pdf, using the COSP satellite simulator of the NCAR CESM model. Differences between the simulated and observed total CFs derived from (e) the default Park, (f) T_pdf, and (g) U_pdf schemes.





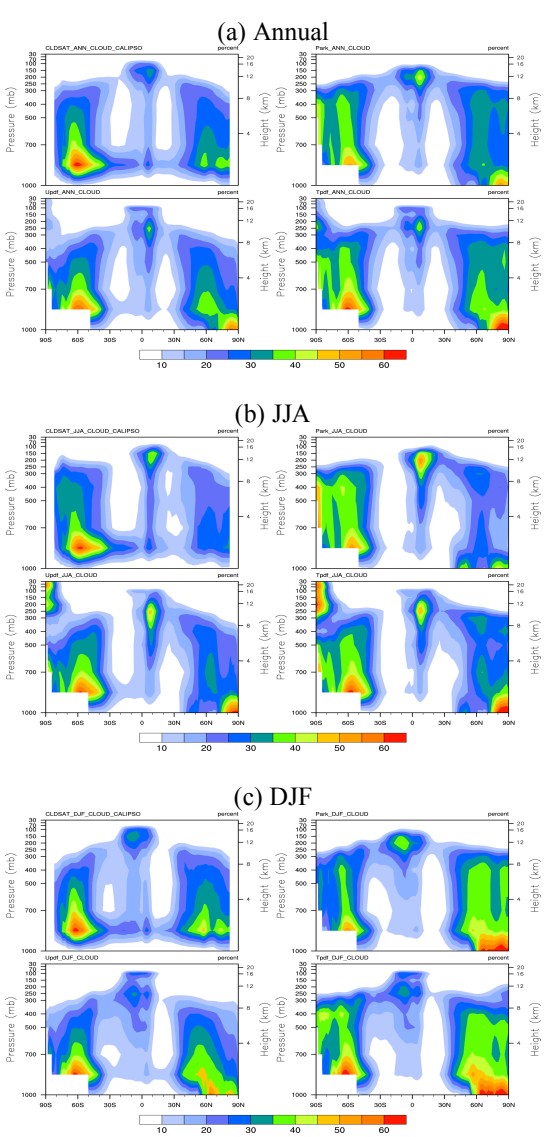

**Figure 6.** Latitude–height cross-sections of (a) annual, (b) June-July-August (JJA), and (c) December-January-February (DJF) mean CFs from CloudSat/CALIPSO data (upper left) and the the Park (upper right), U_pdf (lower left), and T_pdf (lower right) schemes.





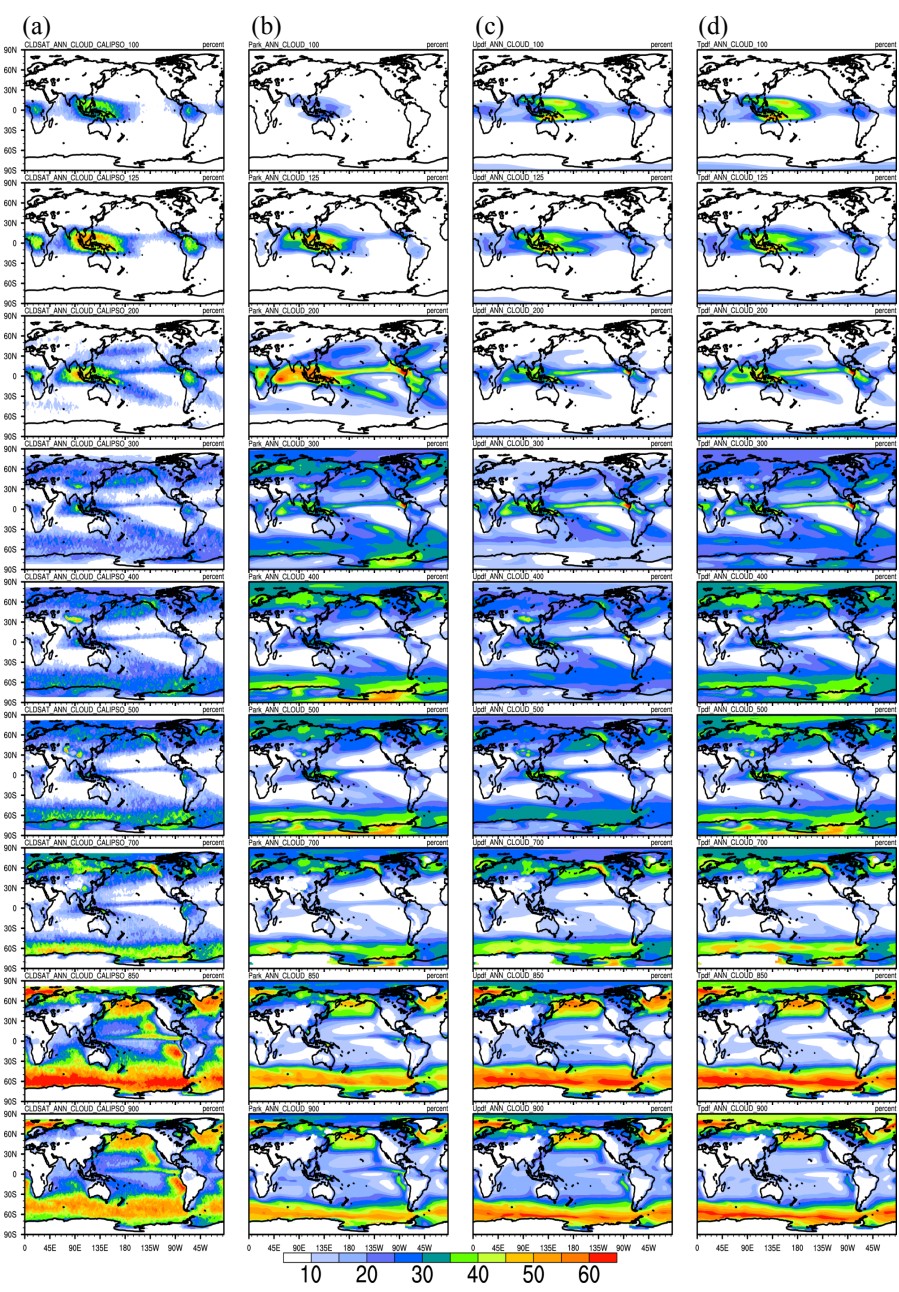

**Figure 7.** CFs at nine pressure levels (one pressure level per row; top to bottom: 100, 125, 200, 300, 400, 500, 700, 850, and 900 mb) from (a) CloudSat/CALIPSO observational data and simulated by (b) the default Park, (c) U_pdf, and (d) T_pdf schemes.

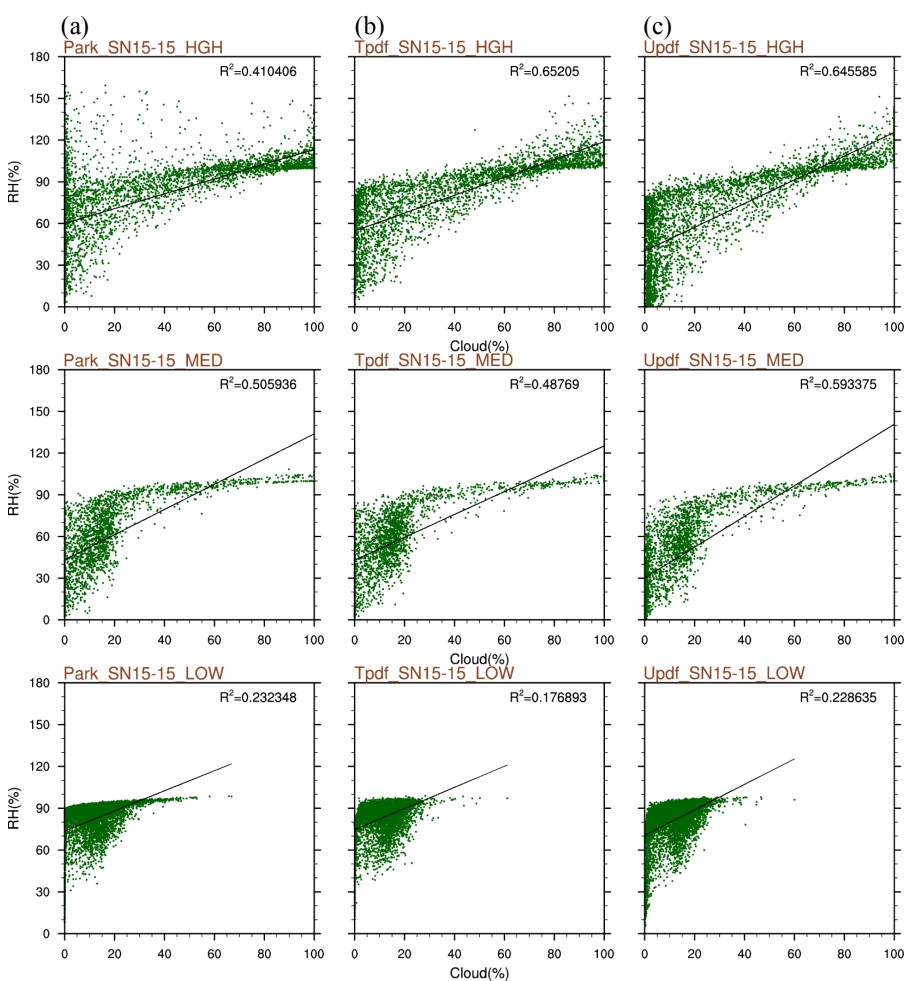

**Figure 8.** CF versus RH at three cloud levels (top to bottom: high, middle, and low clouds) simulated by (a) the default Park, (b) T_pdf, and (c) U_pdf schemes. Daily data of the two grid points ((180° E, 15° N) and (180° E, 15° S)) from 1999 to 2000 are used to generate the scatter plots, and linear regression lines with correlation coefficients are also shown.



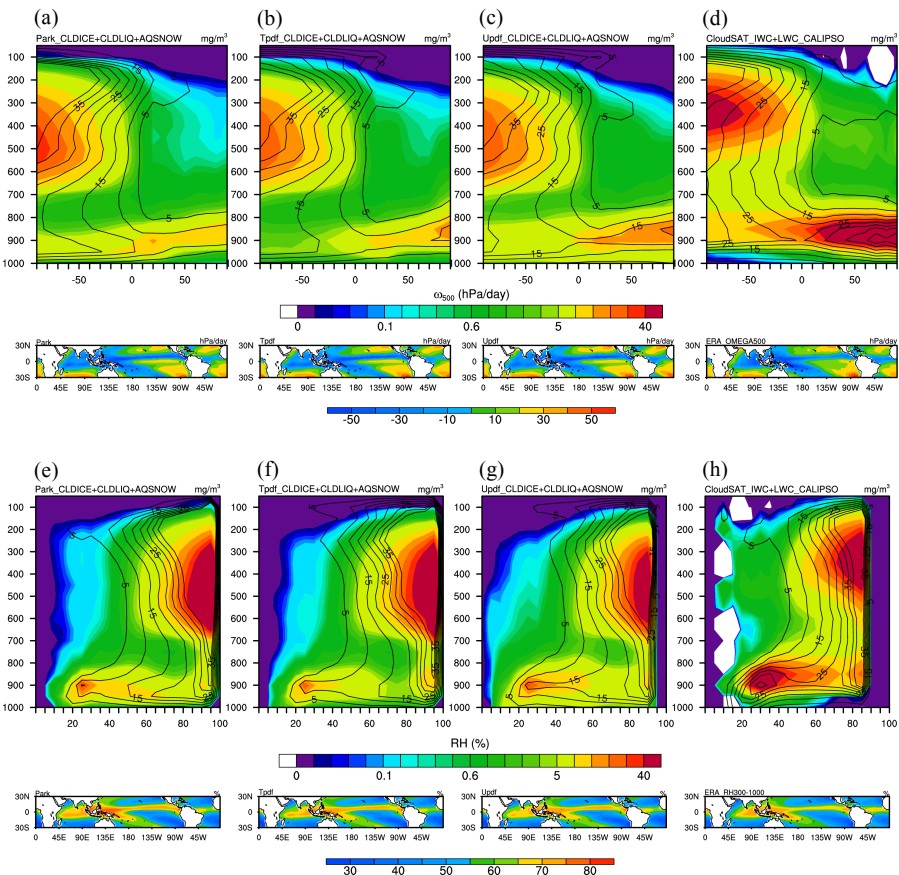

**Figure 9.** Vertical distribution of CF (contour lines) and CWC (colors) as functions of two large-scale parameters: vertical velocity at 500 mb (ω500, upper eight panels) and relative humidity averaged between 300 and 1000 mb (RH300–1000, lower eight panels) for the latitudinal range 30° N–30° S. Columns present simulations by the (a, e) Park, (b, f) T_pdf, and (c, g) U_pdf schemes, and (d, h) observational data from CloudSat/CALIPSO.



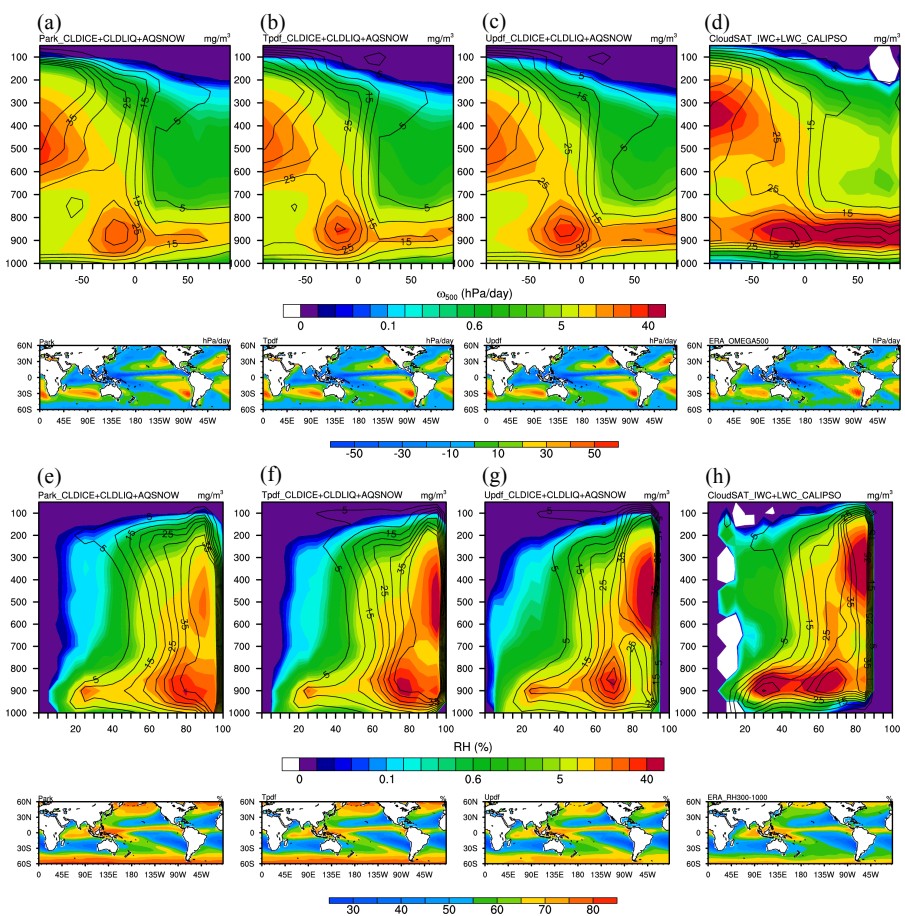

**Figure 10.** Vertical distribution of CF (contour lines) and CWC (colors) as functions of two large-scale parameters: ω500 (upper eight panels) and RH300–1000 (lower eight panels) for the latitudinal range 60° N–60° S. Columns present simulations by the (a, e) Park, (b, f) T_pdf, and (c, g) U_pdf, and (d, h) observational data from CloudSat/CALIPSO.





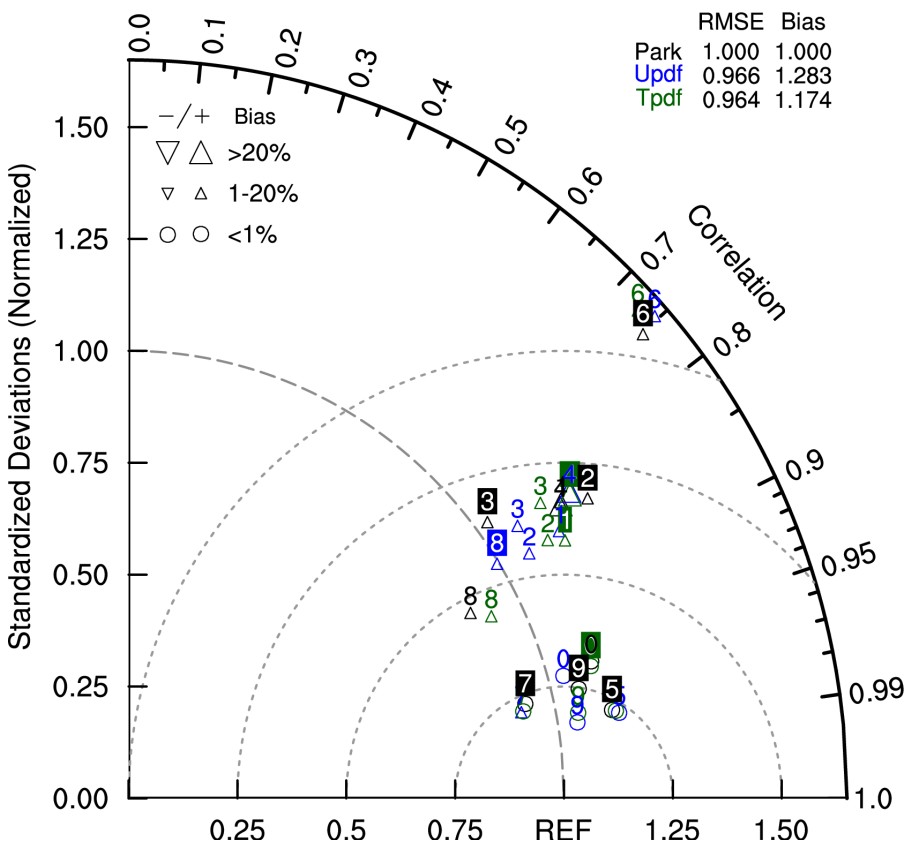

**Figure 11.** Space–time Taylor diagram for the ten climatic parameters simulated by the three macrophysical schemes (Park: black symbols; U_pdf: green; T_pdf: blue) and comparisons of these with the corresponding observational data provided by the atmospheric diagnostic package from the NCAR CESM group. The ten climatic parameters are marked from 0 to 9 where 0 denotes sea level pressure; 1 is SW cloud forcing, 2 is LW cloud forcing, 3 is land rainfall, 4 is ocean rainfall, 5 is land 2-m temperature, 6 is Pacific surface stress, 7 is zonal wind at 300 mb, 8 is relative humidity, and 9 is temperature.





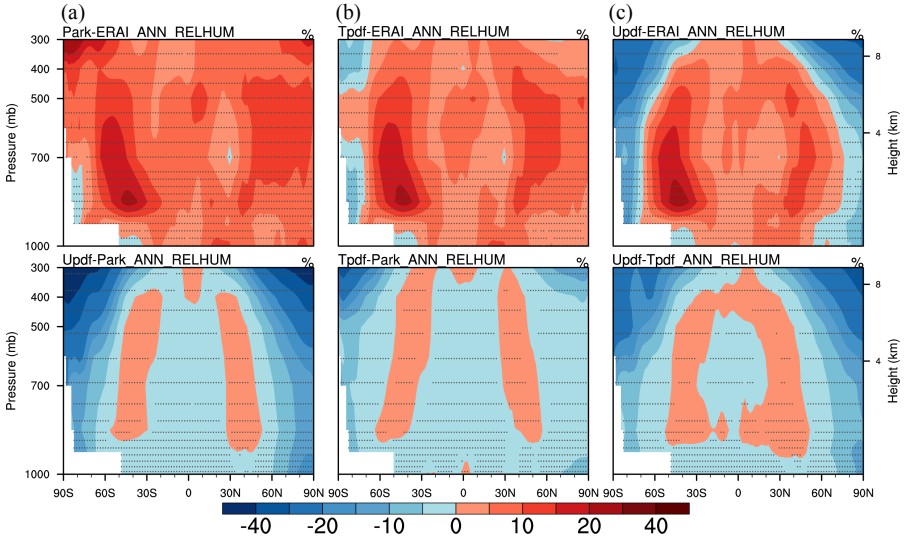

**Figure 12.** Upper row: latitude–pressure cross-sections of differences in relative humidity (RH) between the simulations and ERA-Interim from (a) Park, (b) T_pdf, and (c) U_pdf schemes. Lower row: differences in RH in pair-wise comparisons of the three cloud macrophysical schemes.





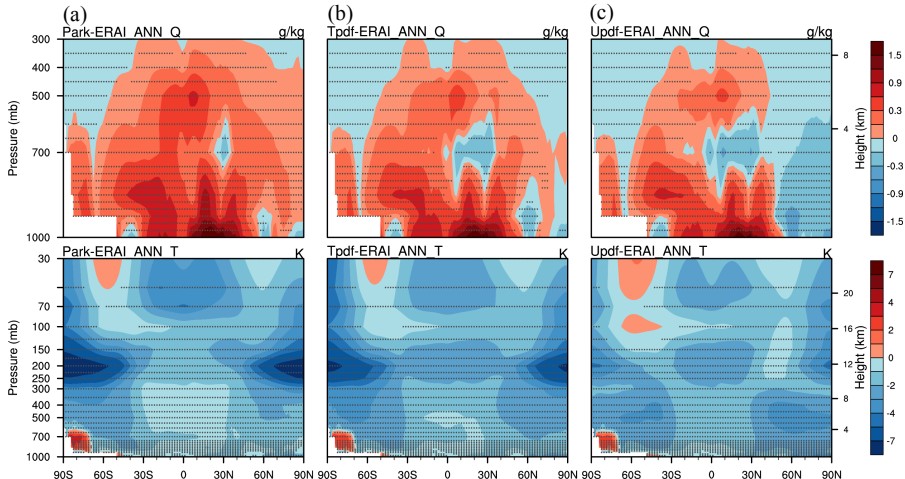

**Figure 13.** Differences in specific humidity (upper row) and air temperature (lower row) between the simulations and ERA-Interim from the (a) Park, (b) T_pdf, and (c) U_pdf schemes.



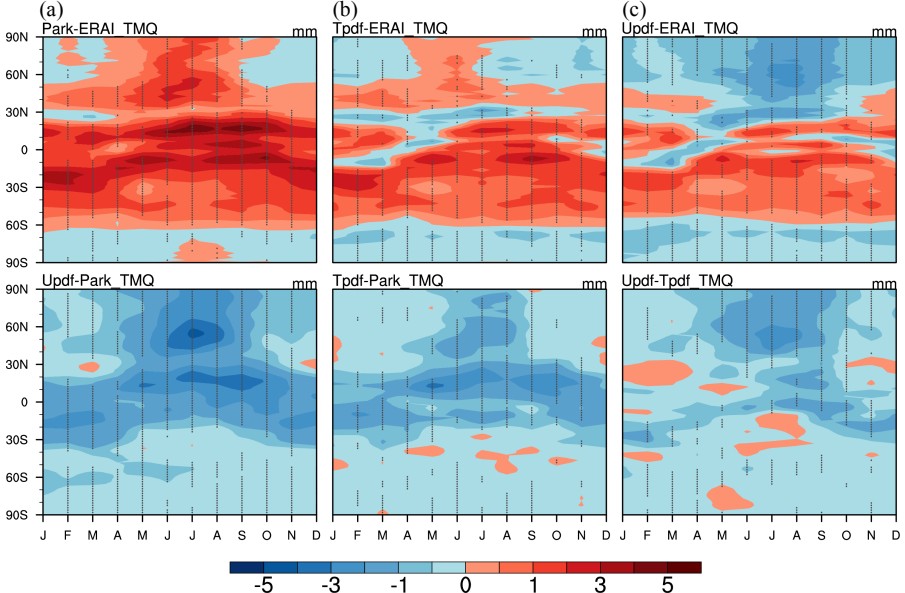

**Figure 14.** Upper row: differences in annual cycles of zonal mean total precipitable water between the three macrophysical schemes and the ERA-Interim data from the (a) Park, (b) T_pdf, and (c) U_pdf schemes. Lower row: differences in annual cycles of total precipitable water in pair-wise comparisons of the three cloud macrophysical schemes.

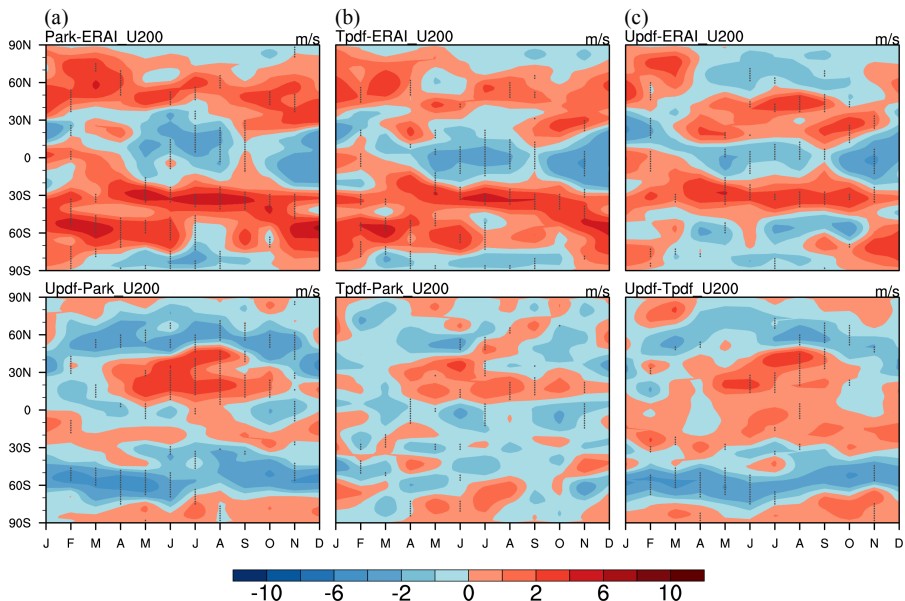

**Figure 15.** Upper row: differences in annual cycles of zonal wind at 200 mb between the three macrophysical schemes and the ERA-Interim data from the (a) Park, (b) T_pdf, and (c) U_pdf schemes. Lower row: differences in annual cycles of zonal wind at 200 mb in pair-wise comparisons of the three cloud macrophysical schemes.





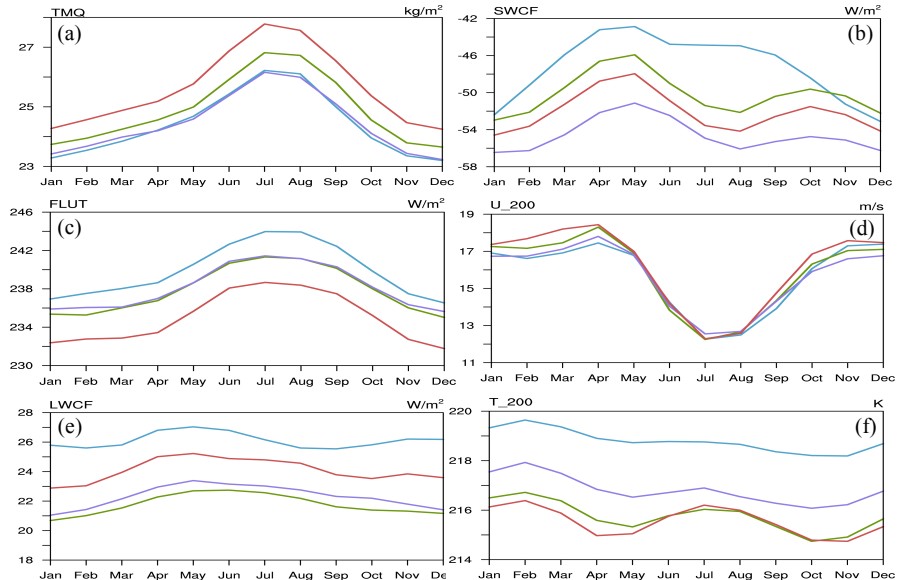

**Figure 16.** Global annual cycles of (a) total precipitable water, (b) shortwave cloud forcing, (c) net longwave flux at the top of the model, (d) zonal wind at 200 mb, (e) longwave cloud forcing, and (f) air temperature at 200 mb. Colored lines represent observational data (blue) and simulations by the Park (red), U_pdf (purple), and T_pdf (green) schemes.



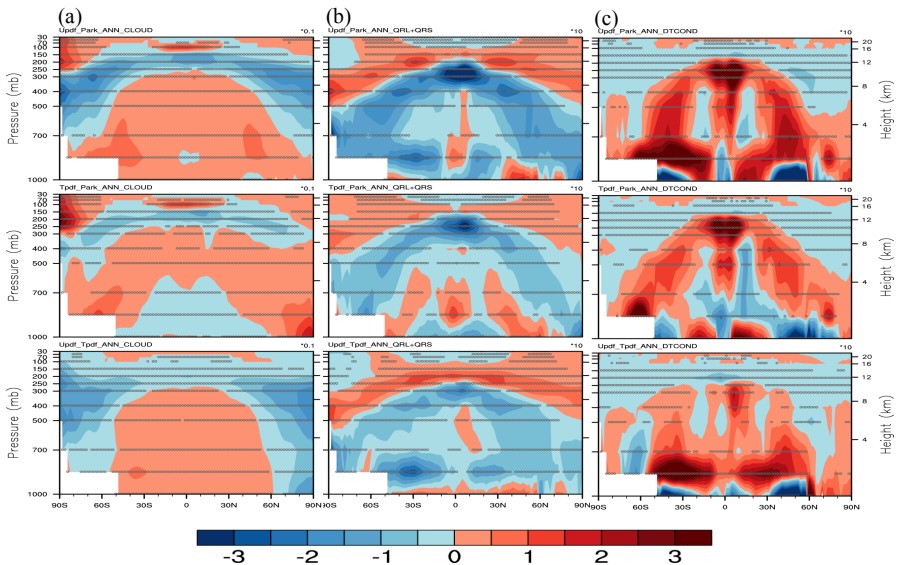

**Figure 17.** Differences in (a) CF, (b) sum of longwave and shortwave heating rates (QRL+QRS), and (c) temperature trends due to moist processes in the NCAR CESM model (DTCOND) in pair-wise comparisons of the three cloud macrophysical schemes. Upper row: U_pdf and Park; middle row: T_pdf and Park; lower row: U_pdf and T_pdf. A statistically significant difference with a confidence level of 95% is represented in the panels by an open circle using Student's t-test.

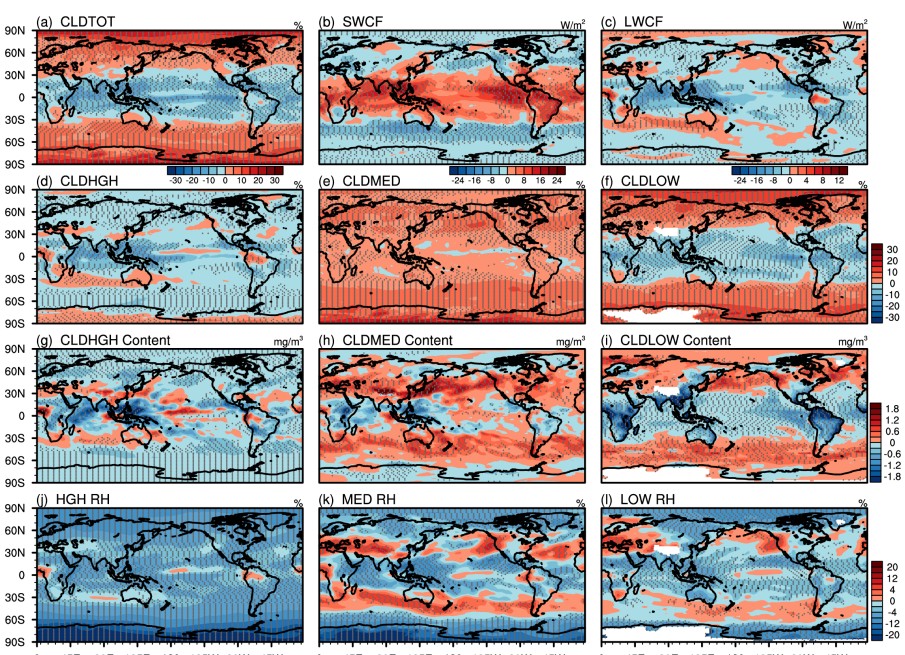

**Figure 18.** Differences in (a) total cloud fraction, (b) short-wave cloud radiative forcing, (c) long-wave cloud radiative forcing, and cloud fraction of (d) high clouds, (e) middle clouds, and (f) low clouds between the T_pdf and default Park schemes. (g–i) As for (d-f) but for total cloud water content at the three cloud levels. (j–l) As for (g–i) except for averaged RH at the three cloud levels.