# Peer review of "GTS v1.0: A Macrophysics Scheme for Climate Models Based on a Probability Density Function Chein-Jung Shiu1,\*, Yi-Chi Wang1, Huang-Hsiung Hsu1, Wei-Ting Chen2, Hua-Lu Pan3, Ruiyu Sun4, Yi-Hsuan Chen5, and Cheng-An Chen"

_Geoscientific Model Development, 2020_

## Referee Comment (RC1) · Anonymous Referee #1 · 8 Sep 2020

The authors developed a new macrophysics scheme (so called, GTS) and showed that the global climate simulated by the GTS scheme has a good quality. This GTS scheme parameterizes both liquid and ice cloud fractions based on the sub-grid distribution of hydrometeor. The GTS uses two different shapes of PDF: one is a symmetric triangular PDF, which is identical to that of GTS's host scheme (the Park scheme), and the other is a uniform PDF. The authors analyzed the performance of the GTS scheme in many ways and showed that the GTS scheme using a uniform PDF (U_pdf) has a better performance than the triangular PDF in most cases. Before being published, I hope the authors address the below comments. My recommendation is to accept the draft with a major revision.

[Figure]

Major comments:

1. Lines 177-182 and others The authors said that "….with the uniform PDF: dC=qs(1-RHc). Therefore, RHc = 1-(dC/qs)." This sentence implies that a uniform PDF with dc corresponds to a symmetric triangular PDF with ãĂŰRHãĂŮ_c of 1-(dc/qs). Although these two distributions have the same dc and RHc as the authors mentioned, they have different variances. The variance of a uniform PDF is (1/3)dCˆ2 but that of a triangular PDF is (1/6)dCˆ2. Instead of using the same half width in two PDFs, isn't it more reasonable to use the same variance for fair comparison ? The authors may repeat the analysis with the same variance.

2. Lines 186-195 This paragraph suggested a formula for the fractional area of ice cloud (bi) as a function of grid-mean water vapor(qv), grid-mean ice condensate(qi), the half-width of PDF, and the saturation specific humidity over ice (qs,i) with a tunable parameter (sup). The authors should provide more detailed explanation on this formula. For example, can the ice cloud fraction be positive when temperature is above 0 degree ? What is 'sup' ? Is the 'qi' used in this formulation the input qi or the qi updated by the GTS scheme or the average of the two ?

3. Lines 219-221 Please provide more explanation on how the T_pdf computes the variable width after clouds are formed. Although the authors mentioned that this variable width is computed using the grid-mean mixing ratio of hydrometeors and the saturation ratio of the environment in Lines 648-654, more detailed explanation is necessary.

4. Lines 298-299 To assess the performance of the scheme, the authors used the ERA-Interim cloud fraction. However, ERA-Interim cloud fraction is not a direct observation but model result. As far as I know, researches do not use ERAI cloud fraction as an observation (in contrast to temperature etc.). The authors should use other data set as "observed" cloud fraction. Also, it may be good to provide some explanations on the sources of the biases in the cloud fraction. Is the bias due to the biases in the

environmental conditions (e.g., environmental relative humidity) or others (incomplete parameterization) under the same environmental conditions ?

5. Lines 290-292 The values of the tunable parameter (i.e., RHc) and the horizontal overlap assumption between liquid and ice cloud fraction, which are used to calculate offline CF in Figure 2, should be explained in detail.

6. Lines 331-334 The authors used the correlation coefficient between RH and CF to evaluate the performance of cloud parameterization. This is very weird: the correlation coefficient only shows the degree of the linear relation between two factors, not the performance of the scheme. In nature, non-linear relationship is likely to exist between RH and CF. Is it fair to say that a higher linear correlation indicates good performance ? I am not sure whether I can agree with the authors' argument.

7. Lines 381-383 and others The GTS scheme parameterizes the large-scale cloud (stratus) fraction in each grid layer. The cloud fraction and associated variables (e.g., cloud radiative forcings) in GCM are also influenced by the parameterizations of convective cloud and vertical cloud overlaps. The authors may want to discuss about this aspect.

8. Lines 648-650 As mentioned above, T_pdf and U_pdf have different variances although they have the same RHc. In other words, the U_pdf uses a wider distribution than the T_pdf. The larger differences between U_pdf and T_pdf compared to the differences between T_pdf and the Park scheme may be due to this difference in the variance.

9. Figure 1 Not all cloud macrophysics schemes assume uniform temperature over the grid. The authors should mention the uniform temperature assumption for the GTS scheme in the main text as well as in the caption of Figure 1.

Technical corrections: 1. Line 176 qt in Eq. 2 should be overlined as it denotes a grid-mean qt.

2. Line 182 use a subscript 'c' in $\delta c$

3. Line 209 The first line of Eq. 6 could be simplified to 1/6 ãĂŰ(1-s_s)ãĂŮˆ3.

---

## Referee Comment (RC2) · Anonymous Referee #2 · 9 Sep 2020

This paper describes a new macrophysics scheme based on PDF and then implements it in CAM5. Two PDF distributions, uniform, and triangular PDF, are tested and compared with the default Park scheme, which uses the triangular distribution. They found the new scheme could simulate clouds and other atmospheric variables better than the default Park scheme. In that, the uniform PDF overall outperforms triangular PDF in many variables. I would recommend publishing this draft after the major revisions listed below.

Major comments:

1. Overall paper organization

1.1 The paper is easy to be followed, but the whole organization could be further improved. I would strongly recommend highlighting the most significant effect of the new scheme on simulations, rather than presenting every perspective of the simulation by comparing with observation with an improved RMSE or correlation. For example, although clouds and radiation both are improved, it is deserved to have more discussion about the reason behind. Another concern is large amounts of tables (Table 1-9) in the main context. Although it is nice to provide overly quantitative results, it might be better to put only the most related tables in the main context and move others in the supplementary.

2. I feel the method part might be not effective for others who want to understand and reproduce the method in-depth. Some other details need to be clarified. Therefore, I suggest the authors to include more details in their method part. Some specific questions are as follows:

2.1 Equation (2), (5) and (6) give the relationship between CF/cloud water and grid mean water condensate and width of the PDF. I cannot exactly follow how the width of PDF is determined based on $q_s$, $q_t$ and $q_l$. For the triangular PDF part, two equations (5) and (6) with two unknown variables could be solved. This is what I can derive from your method. However, equation (2) for CF includes two unknown variables: CF and the width of PDF. If RHc is eliminated, how the weight of PDF is obtained? Maybe you also need to give the equation of cloud water under the uniform assumption. Please clarify your derivation more clearly.

2.2 Moist parameterizations in CAM5 include both convective and stratiform clouds, and they are handled by convection and cloud macrophysics, separately. If my understanding is correct, your new scheme only takes effect on the stratiform cloud fraction. So, the cloud condensate is still handled by the default prognostic cloud condensate scheme, right? My question is: if the new scheme can obtain the good consistency between cloud fraction and condensate, does it still need the consistency check between cloud fraction and condensate to avoid "empty" or "dense" clouds? Please give more

discussion about this in the method part.

2.3 L176: should add one bar for qt because it is a grid-mean value.

2.4 L212-214: "RHc is still required when clouds start to form from a clear region" – What kind of RHc distribution is used for starting the cloud formation? Does it use the same RHc varying with height (low, middle, and high) in the default Park scheme? Please clarify it. A more general question is: will the results be sensitive to the initial given RHc?

3. Some specific questions for results

3.1 I think one of the most important arguments in this paper is the "stronger" relationship between CF and RH while using the GTS scheme than the default Park scheme. However, from the observation signal (Figure 4a, Xie et al), it looks like there is no quite a strong relationship between cloud fraction and RH. My question is: if obs does not show such a feature, why do we need to get this strong relationship in the model? Like Figure 4.

3.2 Radiation balance after using GTS scheme: the radiation balance is quite important for a climate model, and the TOA radiation is very sensitive to modified cloud-related processes. Will the introduced new scheme strongly affect the TOA radiation imbalance? It is necessary to discuss the TOA radiation balance after using the new scheme and overall model climate. Can the imbalance be tuned by other parameters? Will the improvement be "reduced" while reaching a radiation balance?

3.3 Single-column tests:

3.3.1 L327-330: how about the results from U_pdf if it gives a better performance in the offline test?

3.3.2 how do you calculate the correlation between CF and RH in Figure 3? first vertical-integrated CF and RH to get two time series and then correlate them? Please clarify it.

[Figure]

3.3.3 I think there are observation data of cloud fraction and relative humidity for this site. Except for comparing with the default cloud scheme, it will be more informative to compare with observation data. Meanwhile, checking the correlation between CF and RH in observation will give a good reference about which scheme performs best. Furthermore, the observation data could also give a correlation reference between CF and RH.

3.3.4 GTS-Triangular tends to overestimate RH when the cloud fraction is small (Figure 4c). many points are below 40%, and this feature is not shown in observation. why?

3.4 L396-397: why U_pdf results in larger RMSE in JJA and does not perform well? it might be related to a large amount of cloud ice at the upper level in the Arctic in Figure 6b. do you know why there is a second peak of cloud fraction at the upper level of the Arctic? It looks like some "false" clouds are formed at that high level in the GTS scheme. I think more discussions about this bias are necessary since it is highly related to one of the key points in this paper – ice cloud fraction is also calculated by PDF scheme. If it brings such bias, to what extent we should adopt this scheme, especially in the high latitudes?

3.5 Figure 8: I think the scatter plot does not show a great linear correlation between CF and RH, more like exponentials. I am not sure whether I should agree with the author's "higher" linear correlation efficient to evaluate good or bad relationships.

3.6 this paper gives out many statistical results for evaluation, but the reasons behind and the connection between modified clouds and change of other related variables are discussed less. I recommend the authors to add more discussion about this.

3.7 How sensitive the final model's performance to the tuning parameter – sup? From Figure S4, this parameter could reduce the cloud fraction up to 20%, and this will bring a substantial effect on radiation. How the overall model performance will be? Maybe a Taylor plot with different sensitivity simulations could give a good evaluation.

Minor comments:

1. Figure 2: U_pdf performs better at a lower level and has a similar performance with T_pdf at the upper level. what might be the potential reason?

2. L263: in section 3.1, please include the references for each observation data you used and the time period used.

3. L277-279: do you use another CF and CWC data here? It is slightly confusing. I suggest reorganizing this paragraph.

4. L 279: "The methodology from Li et al. (2012) is used to generate gridded data". Are there any specifications for this method to generate gridded data? It might be better to briefly describe the method used here and the reader would not take more time to read the referred paper to figure out what the method is.

5. L388-390: "on the other hand, …." This sentence is confusing. Please consider rephrasing it.

6. Figure 17: what contributes to the stronger DTCOND in U_pdf compared to T_pdf below 700 hPa? The authors might want to add more discussions about it.

Technical:

1. Figures: labels in some figures are too small. Please increase the font size.

2. Figure 9, the lat-lon plot is also too small. Please consider another type of layout for this figure.

3. Figure 17: add the unit of each variable

4. Please consider add legends for line plots.

5. Figure S4: if my understanding is correct, the red solid lines in the upper panel and the lower panel should be from the default Park. However, it looks like they are not the same. Please check it.

---

## Author Comment (AC1) · 7 Oct 2020

Author comments: We would like to thank the two reviewers for their insightful comments, which have greatly helped us to improve the manuscript. Here is the point-by-point reply to reviewers' comments. In the following, the texts with italic font are the reviewer's original comments, and the texts with normal font are authors' responses. The line numbers refer to the clean (no track) version of the revised manuscript. This author comment file, the revised manuscript and the supplementary material can be found in the link: https://drive.google.com/drive/folders/1DGsKjJeuAkDDz4bO2VAGN8wxyPYf011e?usp=sharing

Anonymous Referee #1 The authors developed a new macrophysics scheme (so called, GTS) and showed that the global climate simulated by the GTS scheme has a good quality. This GTS scheme parameterizes both liquid and ice cloud fractions based on the sub-grid distribution of hydrometeor. The GTS uses two different shapes of PDF: one is a symmetric triangular PDF, which is identical to that of GTS's host scheme (the Park scheme), and the other is a uniform PDF. The authors analyzed the performance of the GTS scheme in many ways and showed that the GTS scheme using a uniform PDF (U_pdf) has a better performance than the triangular PDF in most cases. Before being published, I hope the authors address the below comments. My recommendation is to accept the draft with a major revision. Thank you very much for the useful comments. Please kindly find our responses as listed below.

Major comments: 1. Lines 177-182 and others The authors said that ". . ..with the uniform PDF: dC=qs(1- RHc). Therefore, RHc = 1-(dC/qs)." This sentence implies that a uniform PDF with dc corresponds to asymmetric triangular PDF with aǏČA ÌĘU ÌŃRHaǏČA ÌĘU ÌŁ_cof1-(dc/qs). Although these two distributions have the same dc and RHc as the authors mentioned, they have different variances. The variance of a uniform PDF is (1/3)dCËĘ2 but that of a triangular PDF is (1/6)dCËĘ2. Instead of using the same half width in two PDFs, isn't it more reasonable to use the same variance for fair comparison? The authors may repeat the analysis with the same variance. Thank you for the comments. The half width of uniform or triangular PDF of GTS scheme is not determined by RHc. We have added the relevant equations to clarify this issue in Section 2.1 via using uniform distribution of GTS as an example as shown in lines 187-194 of the revised manuscript. Also, we have added the Appendix A in the revised manuscript to demonstrate the detailed derivations of equations (8) and (9) which are used in the T_pdf of GTS scheme to calculate $\delta$ and cloud fraction. Please find the descriptions as follows. With uniform PDF as denoted in Figure 1 (a), the liquid cloud fraction (bl) and grid-mean cloud-liquid mixing ratio (($q\_l$ ) ǏĔ) can be integrated as follows: b_l=∫ _($q\_s$)∞▒ãŰŰP(q_t )dq_t ãŰŮ=1/2$\delta$ ($Âŕ$(q_l )+(q_v ) ǏĔ+$\delta$-q_s ), (4)

[Figure]

and: $\hat{r}(q\_l) = \int\_{(q\_s}$⊖∞$\hat{a}\text{ŰŠ}\tilde{a}\text{ĂŰ}(q\_t-q\_s)P(q\_t)dq\_t$ $\tilde{a}\text{ĂŮ}=1/4\delta$ $(\hat{r}(q\_t)+\delta-q\_s)$. (5) Given $(q\_l)$ ÌĚ, $(q\_v)$ ÌĚ, and $q\_s$, the width of uniform PDF can be determined as follows: $\delta=(\sqrt{((q\_l)\text{ ÌĚ})}+\sqrt{(q\_s-(q\_v)\text{ ÌĚ})})^2$. (6) Therefore, we can calculate the liquid cloud fraction from equation (4).

2. Lines 186-195 This paragraph suggested a formula for the fractional area of ice cloud (bi) as a function of grid-mean water vapor(qv), grid-mean ice condensate(qi), the half-width of PDF, and the saturation specific humidity over ice (qs,i) with a tunable parameter (sup). The authors should provide more detailed explanation on this formula. For example, can the ice cloud fraction be positive when temperature is above 0 degree What is 'sup'? Is the 'qi' used in this formulation the input qi or the qi updated by the GTS scheme or the average of the two? Thank you for the suggestion. Yes. We have added more explanations in lines 200-205: "In equation (7), q_si is multiplied by a supersaturation factor (sup) to account for the situation in which rapid saturation adjustment is not reached for cloud ice. In the present version of the GTS scheme, sup is temporarily assumed to be 1.0. Sensitivity tests regarding sup will be discussed in Section 5.7. Values of $\hat{r}(q\_i)$ and $\hat{r}(q\_v)$ used to calculate equation (7) are the updated state variables before calling the cloud macrophysics process."

3. Lines 219-221 Please provide more explanation on how the T_pdf computes the variable width after clouds are formed. Although the authors mentioned that this variable width is computed using the grid-mean mixing ratio of hydrometeors and the saturation ratio of the environment in Lines 648-654, more detailed explanation is necessary. Thank you for the suggestion. Yes. We have added more explanation on how U_pdf or T_pdf computes the variable width after clouds are formed. Explicit equations for calculating cloud fraction and half width are added in Section 2 both for the U_pdf and T_pdf of the GTS schemes. Please kindly find our response to the major comment #1.

4. Lines 298-299 To assess the performance of the scheme, the authors used the ERA- Interim cloud fraction. However, ERA-Interim cloud fraction is not a direct observation but model result. As far as I know, researches do not use ERAI cloud fraction as an observation (in contrast to temperature etc.). The authors should use other data set as "observed" cloud fraction. Also, it may be good to provide some explanations on the sources of the biases in the cloud fraction. Is the bias due to the biases in the environmental conditions (e.g., environmental relative humidity) or others (incomplete parameterization) under the same environmental conditions? Thanks for pointing out this concern. We realized that ERA-Interim cloud fraction is also obtained from the global model simulation and it cannot be used as "observed" cloud fraction. Purpose of the offline calculations in the paper is to evaluate the impact of assumptions of CF distributions for the RH- and PDF-based schemes via using a balanced atmospheric state (the cloud liquid mixing ratio ((q_l ) ÌĚ), water vapor mixing ratio ((q_v ) ÌĚ), and air temperature (to calculate (q_sl ) ÌĚ)) provided by the reanalysis data. The cloud fraction of ERA-Interim is just used as a reference for checking occurrence of clouds and CF distributions. We actually used the CF of CloudSAT/CALIPSO as observations to access the performance of the GTS scheme as discussed in section 5 and shown in Figures 5–9. The biases are likely to be caused by the assumptions of the parameterizations.

5. Lines 290-292 The values of the tunable parameter (i.e., RHc) and the horizontal overlap assumption between liquid and ice cloud fraction, which are used to calculate offline CF in Figure 2, should be explained in detail. Thank you for the suggestion. Yes. We have added more descriptions to explain the details of offline calculations as shown in lines 327-338: "Using the U_pdf of GTS scheme as an example to elaborate on the details of calculation procedures, we simply obtain the cloud liquid mixing ratio ((q_l ) ÌĚ), water vapor mixing ratio ((q_v ) ÌĚ), and air temperature (to calculate (q_sl ) ÌĚ) from the ERA-Interim as input variables to calculate the liquid CF via using equations (6) and (4) when (q_l ) ÌĚ is greater than 10-10 (kg kg-1). When (q_l ) ÌĚ is smaller than 10-10 (kg kg-1) and if RH > RHc, CFs are calculated based on equation (3) and the liquid CF parameterization of Sundqvist et al. [1989] and if RH < RHc, CFs are equal to zero. Ice CFs are calculated similarly as those of liquid CFs but using equation (7),

$\hat{r}(q\_i)$, $\hat{r}(q\_si)$, and sup = 1.0. Procedures for calculating CFs diagnosed by the T_pdf of GTS scheme are similar to those of U_pdf but using equation set of triangular PDF. Values of RHc used in the U_pdf and T_pdf of GTS schemes are assumed to be 0.8 and height-independent. Maximum overlapping assumption is used to calculate the horizontal overlap between the liquid CF and ice CF.".

6. Lines 331-334 The authors used the correlation coefficient between RH and CF to evaluate the performance of cloud parameterization. This is very weird: the correlation coefficient only shows the degree of the linear relation between two factors, not the performance of the scheme. In nature, non-linear relationship is likely to exist between RH and CF. Is it fair to say that a higher linear correlation indicates good performance? I am not sure whether I can agree with the authors' argument. Thank you for the useful comments. We totally agree with the two reviewers' comments (reviewer #2 also gives similar comments). As reviewer's comments, in nature, non-linear relationship is likely to exist between RH and CF as shown in the observational results during the TWP-ICE experiment. We have removed those sentences and discussions about the increase of linear correlation between RH and CF in Section 4 (i.e., the single column model part in the manuscript). Similarly, we also removed all those wrong discussions and relevant figures in Section 5 (i.e., the global simulation part). We have rewritten the whole Section 4 to avoid any potential misunderstanding. Please kindly find the completed modifications in the revised manuscript. For example, we have modified the descriptions in lines 404-410: "Notably, such pair comparisons (i.e., T_pdf with Slingo cloud ice fraction scheme vs. T_pdf and vs. Park) only reveal the important features of the GTS scheme, such as how variations in liquid CF are better correlated with changes in RH of the GCM grids when compared to that of the default cloud macrophysics scheme. In fact, such high correlations between CF and RH seen in the GTS and Park schemes are not consistent with those of observations as shown in Figure 3(a), suggesting that, in nature, CF and RH is likely to be non-linear." Another example in lines 411-417: "Admittedly, it is not easy to directly use the observational CF of TWP-ICE field campaign to evaluate the performances of stratiform cloud macrophysics schemes

in the SCAM simulations due to the co-existing of other CF types determined by the deep and shallow convective schemes as well as cloud overlapping treatments in both horizontal and vertical directions. As expected, correlation coefficients between the simulated and observed CFs are not high and their values do not differ a lot among the five cloud macrophysics schemes (Table S1)." We have also modified similar concerns in "Abstract" and "Conclusions" part in lines 49-51: "Via single-column model results, the new approach simulates the cloud fraction (CF)–RH distributions closer to those of the observations when compared to those of the default CAM5.3 scheme." and in lines 838-843: "In a single-column model setup, we noticed, according to the pair-wise comparisons shown and discussed in Figures 3 and 4, the use of PDF-based treatments for parameterizing both liquid and ice CFs in the GTS schemes contributed to the CF-RH distributions. The GTS schemes simulated the CF-RH distributions closer to those of the observational results compared to the default scheme of CAM5.3.".

7. Lines 381-383 and others The GTS scheme parameterizes the large-scale cloud (stratus) fraction in each grid layer. The cloud fraction and associated variables (e.g., cloud radiative forcings) in GCM are also influenced by the parameterizations of convective cloud and vertical cloud overlaps. The authors may want to discuss about this aspect. Thank you for the suggestion. Yes. We have added descriptions related to this concern in lines 441-444: "Notably, the following comparisons for the CF and associated variables are not only affected by the changes in the cloud macrophysics but also contributed by the deep and shallow convective schemes as well as cloud overlapping assumptions in the horizontal and vertical directions."

8. Lines 648-650 As mentioned above, T_pdf and U_pdf have different variances although they have the same RHc. In other words, the U_pdf uses a wider distribution than the T_pdf. The larger differences between U_pdf and T_pdf compared to the differences between T_pdf and the Park scheme may be due to this difference in the variance. Thanks for the comments. We apologized for the confusing organization of old version of Section 2.1. As mentioned in our response to the major comments #1,

the half width of U_pdf or T_pdf of GTS scheme is variable. RHc is no longer used once clouds formed. We have clarified this in Section 2.1 of the revised manuscript.

9. Figure 1 Not all cloud macrophysics schemes assume uniform temperature over the grid. The authors should mention the uniform temperature assumption for the GTS scheme in the main text as well as in the caption of Figure 1. Thank you for reminding this issue. Yes. We have added the relevant descriptions in the main text in line 186: "Please note that uniform temperature is assumed over the grid for the GTS scheme." and in the caption of Figure 1: "Please note that uniform temperature assumption is used for the GTS cloud macrophysics.".

Technical corrections: 1. Line 176 qt in Eq. 2 should be overlined as it denotes a grid-mean qt. Thanks. Corrected.

2. Line 182 use a subscript 'c' in $\delta c$ Thanks. Corrected.

3.Line209ThefirstlineofEq.6couldbesimplifiedto1/6aÌČA ÌĘU ÌŃ(1-s_s)aÌČA ÌĘU ÌŁĖĘ3. Thanks for the suggestion. This equation is somehow misplaced and wrong in the old version. We have corrected it.

 

Anonymous Referee #2 This paper describes a new macrophysics scheme based on PDF and then implements it in CAM5. Two PDF distributions, uniform, and triangular PDF, are tested and compared with the default Park scheme, which uses the triangular distribution. They found the new scheme could simulate clouds and other atmospheric variables better than the default Park scheme. In that, the uniform PDF overall outperforms triangular PDF in many variables. I would recommend publishing this draft after the major revisions listed below. Thank you very much for the useful comments. Please kindly find our responses as listed below.

Major comments:

1. Overall paper organization 1.1 The paper is easy to be followed, but the whole

organization could be further improved. I would strongly recommend highlighting the most significant effect of the new scheme on simulations, rather than presenting every perspective of the simulation by comparing with observation with an improved RMSE or correlation. For example, although clouds and radiation both are improved, it is deserved to have more discussion about the reason behind. Another concern is large amounts of tables (Table 1-9) in the main context. Although it is nice to provide overly quantitative results, it might be better to put only the most related tables in the main context and move others in the supplementary. Thank you for the useful comments. Yes. We have tried to add some more discussions about the reason behind as possible. Admittedly, it is not easy to disentangle the relationship between causes and effects resulted from the usage of the GTS scheme in the global simulations as we discussed in Section 5.6 of the revised manuscript. We have removed 5 Tables to the supplementary material and kept only four Tables to show the performance of GTS schemes.

2. I feel the method part might be not effective for others who want to understand and reproduce the method in-depth. Some other details need to be clarified. Therefore, I suggest the authors to include more details in their method part. Some specific questions are as follows: 2.1 Equation (2), (5) and (6) give the relationship between CF/cloud water and grid mean water condensate and width of the PDF. I cannot exactly follow how the width of PDF is determined based on qs, qt and ql. For the triangular PDF part, two equations (5) and (6) with two unknown variables could be solved. This is what I can derive from your method. However, equation (2) for CF includes two unknown variables: CF and the width of PDF. If RHc is eliminated, how the weight of PDF is obtained? Maybe you also need to give the equation of cloud water under the uniform assumption. Please clarify your derivation more clearly. Thank you for the comments. We apologized for the confusing organization of Section 2.1 in the old version of manuscript. This concern is also pointed out by the reviewer #1. Yes. We have added the relevant equations to clarify this issue in Section 2.1 via using uniform distribution of GTS as an example as shown in lines 187-194 of the revised manuscript.

Also, we have added the Appendix A in the revised manuscript to demonstrate the detailed derivations of equations (8) and (9) which are used in the T_pdf of GTS scheme to calculate $\delta$ and cloud fraction. Please find the descriptions as follows. With uniform PDF as denoted in Figure 1 (a), the liquid cloud fraction (bl) and grid-mean cloud-liquid mixing ratio (($q\_l$) ÌĚ) can be integrated as follows: b_l=$\int$ _($q\_s$)⊖∞▒ãĂŰP($q\_t$ )dq_t ãĂŮ=1/2$\delta$ (Âŕ(q_l )+(q_v ) ÌĚ+$\delta$-q_s ), (4) and: Âŕ(q_l )=$\int$ _($q\_s$)⊖∞▒ãĂŰ(q_t-q_s)P($q\_t$ )dq_t ãĂŮ=1/4$\delta$ (Âŕ(q_t )+$\delta$-q_s ). (5) Given (q_l ) ÌĚ, (q_v ) ÌĚ, and q_s, the width of uniform PDF can be determined as follows: $\delta$=($\sqrt{}$((q_l ) ÌĚ )+$\sqrt{}$(q_s-(q_v ) ÌĚ ))ˆ2. (6) Therefore, we can calculate the liquid cloud fraction from equation (4).

2.2 Moist parameterizations in CAM5 include both convective and stratiform clouds, and they are handled by convection and cloud macrophysics, separately. If my understanding is correct, your new scheme only takes effect on the stratiform cloud fraction. So, the cloud condensate is still handled by the default prognostic cloud condensate scheme, right? My question is: if the new scheme can obtain the good consistency between cloud fraction and condensate, does it still need the consistency check between cloud fraction and condensate to avoid "empty" or "dense" clouds? Please give more discussion about this in the method part. Thank you for the comment. Yes. Following reviewer #2's suggestions, we have added more discussions in lines 226-232: "The GTS scheme still uses the default prognostic scheme for calculating cloud condensates [Park et al., 2014] and it takes effects only on the stratiform CFs. Although the GTS scheme is presumed to have good consistency between CF and condensates, the consistency check subroutines of the Park scheme are still kept in the GTS scheme to avoid "empty" and "dense" clouds due to the usage of Park scheme for calculating cloud condensates and the GTS schemes still need RHc when clouds start to form.".

2.3 L176: should add one bar for qt because it is a grid-mean value. Yes. Corrected as suggested.

2.4 L212-214: "RHc is still required when clouds start to form from a clear region" – What kind of RHc distribution is used for starting the cloud formation? Does it use

the same RHc varying with height (low, middle, and high) in the default Park scheme? Please clarify it. A more general question is: will the results be sensitive to the initial given RHc? Thanks for the comment. Yes. We have added the assumption of RHc used in the GTS scheme in lines 225-226: "To simplify the cloud macrophysics parameterization, value of RHc in GTS scheme is assumed to be 0.8 instead of RHc varying with height in the default Park scheme.". We have also discussed the sensitivity issue of RHc in an added new subsection in lines 813-823: "In addition, RHc of cloud macrophysics parameterizations are frequently used to tune the radiation balance issue of coupled GCMs. As mentioned in section 2.1, although RHc is no longer used once clouds formed in the GTS schemes, the GTS schemes still need RHc when clouds start to form. RHc is assumed to be 0.8 and height-independent in this study. Our past tuning experiences suggest that tuning RHc of GTS scheme could moderately alter the net radiation flux at TOA of coupled global simulations. For example, the net radiation fluxes at TOA are –0.61 and –0.23 W m-2 for RHc = 0.83 and RHc = 0.85, respectively, in TaiESM tuning work using T_pdf of GTS scheme. Therefore, RHc in the GTS scheme can be one of the parameters for tuning GCMs. Moreover, height-dependent RHc as that of the Park cloud macrophysics scheme can be considered to tune the TOA radiation balance.".

3. Some specific questions for results 3.1 I think one of the most important arguments in this paper is the "stronger" relation- ship between CF and RH while using the GTS scheme than the default Park scheme. However, from the observation signal (Figure 4a, Xie et al), it looks like there is no quite a strong relationship between cloud fraction and RH. My question is: if obs does not show such a feature, why do we need to get this strong relationship in the model? Like Figure 4. Thank you for the useful comments. We totally agree with the two reviewers' comments (reviewer #1 also give similar comments). As reviewer's comments, in nature, non-linear relationship is likely to exist between RH and CF as shown in the observational results during the TWP-ICE experiment. We have removed those sentences and discussions about the increase of linear correlation between RH and CF in Section 4 (i.e., the single column model part

in the manuscript). Similarly, we also removed all those wrong discussions and relevant figures in Section 5 (i.e., the global simulation part). We have rewritten the whole Section 4 to avoid any potential misunderstanding. Please kindly find the completed modifications in the revised manuscript. For example, we have modified the descriptions in lines 404-410: "Notably, such pair comparisons (i.e., T_pdf with Slingo cloud ice fraction scheme vs. T_pdf and vs. Park) only reveal the important features of the GTS scheme, such as how variations in liquid CF are better correlated with changes in RH of the GCM grids when compared to that of the default cloud macrophysics scheme. In fact, such high correlations between CF and RH seen in the GTS and Park schemes are not consistent with those of observations as shown in Figure 3(a), suggesting that, in nature, CF and RH is likely to be non-linear." Another example in lines 411-417: "Admittedly, it is not easy to directly use the observational CF of TWP-ICE field campaign to evaluate the performances of stratiform cloud macrophysics schemes in the SCAM simulations due to the co-existing of other CF types determined by the deep and shallow convective schemes as well as cloud overlapping treatments in both horizontal and vertical directions. As expected, correlation coefficients between the simulated and observed CFs are not high and their values do not differ a lot among the five cloud macrophysics schemes (Table S1)." We have also modified similar concerns in "Abstract" and "Conclusions" part in lines 49-51: "Via single-column model results, the new approach simulates the cloud fraction (CF)–RH distributions closer to those of the observations when compared to those of the default CAM5.3 scheme." and in lines 838-843: "In a single-column model setup, we noticed, according to the pair-wise comparisons shown and discussed in Figures 3 and 4, the use of PDF-based treatments for parameterizing both liquid and ice CFs in the GTS schemes contributed to the CF-RH distributions. The GTS schemes simulated the CF-RH distributions closer to those of the observational results compared to the default scheme of CAM5.3.".

3.2 Radiation balance after using GTS scheme: the radiation balance is quite important for a climate model, and the TOA radiation is very sensitive to modified cloud-related processes. Will the introduced new scheme strongly affect the TOA radiation

imbalance? It is necessary to discuss the TOA radiation balance after using the new scheme and overall model climate. Can the imbalance be tuned by other parameters? Will the improvement be "reduced" while reaching a radiation balance? Thank you for the suggestions. As per reviewer' suggestion, we have added a new subsection to discuss this concern in lines 784-823: "c. Tuning parameters of the GTS scheme The top of atmosphere (TOA) radiation balance is very important for a coupled climate model and modifying cloud-related physical parameterizations can significantly alter the TOA radiation balance. It is thus worth comparing the difference in TOA radiation flux between the GTS and the default Park schemes as listed in Table 4. It turns out that the net TOA radiation of T_pdf is smaller than that of the Park scheme by 0.93 W m-2. In contrast, the net TOA radiation of U_pdf is smaller than that of the Park scheme by 5.24 W m-2. We can expect that utilizing U_pdf of the GTS scheme will introduce much stronger TOA radiation imbalance compared to T_pdf of the GTS scheme in present physical parameterization framework of NCAR CESM 1.2.2. Our past experiences in tuning GCMs also show that implementing strong tuning sometimes will indeed offset the improvements resulted from physical parameterizations with less tuning. In fact, to avoid the situation, we used the T_pdf of GTS scheme (with tuning as discussed below) as the stratiform cloud macrophysics scheme of TaiESM model for participating the CMIP6 project [Lee et al., 2020]. As mentioned in the previous subsection, the sup value can be tuned and CF profiles would be modified accordingly as shown in Figure S5. It is thus worth discussing the sensitivity of tuning parameters of the GTS scheme and whether such tuning would affect overall model performance. It is interesting to note that, although significant changes in CF profiles (Figure S5), SWCF, and LWCF (Table S8) between sup = 1.0 and sup = 1.05 are shown, differences in net radiation at the top of model (RESTOM) between sup = 1.0 and sup = 1.05 are only about 0.6 to 0.7 W m-2 for the GTS schemes (Table S8). Such outcome suggests that possible compensating effects exist between changes in SWCF and LWCF associated with cloud overlapping. One could expect that, despite relatively smaller changes in RESTOM, significant changes in SWCF and LWCF between sup = 1.0 and

sup = 1.05 could potentially affect the overall performance of GCMs. Comparisons of Taylor diagrams and biases confirm this (Figures S6 and S7, Table S9). Notably, sup here is assumed to be a constant and height-independent. Further height-dependent tuning can be tested. In addition, RHc of cloud macrophysics parameterizations are frequently used to tune the radiation balance issue of coupled GCMs. As mentioned in section 2.1, although RHc is no longer used once clouds formed in the GTS schemes, the GTS schemes still need RHc when clouds start to form. RHc is assumed to be 0.8 and height-independent in this study. Our past tuning experiences suggest that tuning RHc of GTS scheme could moderately alter the net radiation flux at TOA of coupled global simulations. For example, the net radiation fluxes at TOA are –0.61 and –0.23 W m-2 for RHc = 0.83 and RHc = 0.85, respectively, in TaiESM tuning work using T_pdf of GTS scheme. Therefore, RHc in the GTS scheme can be one of the parameters for tuning GCMs. Moreover, height-dependent RHc as that of the Park cloud macrophysics scheme can be considered to tune the TOA radiation balance.".

3.3 Single-column tests: 3.3.1 L327-330: how about the results from U_pdf if it gives a better performance in the offline test? Thanks for the suggestion. Yes. We have added the results from U_pdf in Section 4 and discussed accordingly. It turns out that U_pdf does not give a better performance as what seen in the global simulations. We have discussed about this in lines 408-417: "In fact, such high correlations between CF and RH seen in the GTS and Park schemes are not consistent with those of observations as shown in Figure 3(a), suggesting that, in nature, CF and RH is likely to be non-linear. Admittedly, it is not easy to directly use the observational CF of TWP-ICE field campaign to evaluate the performances of stratiform cloud macrophysics schemes in the SCAM simulations due to the co-existing of other CF types determined by the deep and shallow convective schemes as well as cloud overlapping treatments in both horizontal and vertical directions. As expected, correlation coefficients between the simulated and observed CFs are not high and their values do not differ a lot among the five cloud macrophysics schemes (Table S1).".

3.3.2 how do you calculate the correlation between CF and RH in Figure 3? first vertical-integrated CF and RH to get two time series and then correlate them? Please clarify it. Thanks for the comment. We actually calculated the pressure-time cross-section pattern correlation between CF and RH.

3.3.3 I think there are observation data of cloud fraction and relative humidity for this site. Except for comparing with the default cloud scheme, it will be more informative to compare with observation data. Meanwhile, checking the correlation between CF and RH in observation will give a good reference about which scheme performs best. Furthermore, the observation data could also give a correlation reference between CF and RH. Thank you for the useful comments. Yes, We have added the observational results to Figures 3 and 4 for the single column model comparisons and discussed accordingly. Part of relevant discussion can be seen in lines 408-432: "In fact, such high correlations between CF and RH seen in the GTS and Park schemes are not consistent with those of observations as shown in Figure 3(a), suggesting that, in nature, CF and RH is likely to be non-linear. Admittedly, it is not easy to directly use the observational CF of TWP-ICE field campaign to evaluate the performances of stratiform cloud macrophysics schemes in the SCAM simulations due to the co-existing of other CF types determined by the deep and shallow convective schemes as well as cloud overlapping treatments in both horizontal and vertical directions. As expected, correlation coefficients between the simulated and observed CFs are not high and their values do not differ a lot among the five cloud macrophysics schemes (Table S1). To minimize possible interference from deep and shallow convective CFs, we picked up the stratiform cloud-dominated levels and time period to examine the CF-RH distributions. Figure 4 shows scatter plots of RH and CF between 50 and 300 hPa determined from observations [Xie et al., 2010] and simulated by models run for the suppressed monsoon period from the TWP-ICE case. It turns out that the CF-RH distributions simulated by the GTS schemes (Figures 4(c) and 4(f)) are closer to those of the observational results (Figure 4(a)) except under more overcast conditions (i.e., RH > 70% and RH > 110%). In contrast, the CF-RH distributions simulated by the Park scheme are much

less consistent with those of observations (Figures 4(d) vs. 4(a)). On the other hand, by excluding PDF-based treatment for the cloud ice fraction in the GTS scheme, a more obvious spread in the CF-RH distribution is produced (comparing Figures 4(b) and 4(c) or Figures 4(e) and 4(f)). In other words, the comparisons shown in Figure 4 suggest that applying a PDF-based treatment for both liquid and ice CF parameterizations can simulate the CF-RH distributions in better agreements with the observational results.".

3.3.4 GTS-Triangular tends to overestimate RH when the cloud fraction is small (Figure 4c). many points are below 40%, and this feature is not shown in observation. why? Thanks for the comments. We suggest that it is probably related to the usage of cloud hydrometeor mixing ratio to calculate cloud fraction in the GTS scheme. When in low RH environments, GTS schemes still can have small CF if cloud liquid or cloud ice exists.

3.4 L396-397: why U_pdf results in larger RMSE in JJA and does not perform well? it might be related to a large amount of cloud ice at the upper level in the Arctic in Figure 6b. do you know why there is a second peak of cloud fraction at the upper level of the Arctic? It looks like some "false" clouds are formed at that high level in the GTS scheme. I think more discussions about this bias are necessary since it is highly related to one of the key points in this paper – ice cloud fraction is also calculated by PDF scheme. If it brings such bias, to what extent we should adopt this scheme, especially in the high latitudes? Thank you for noticing this concern. Yes. The U_pdf results show larger RMSE in JJA and it could be related to the "false" clouds as suggested. We have added more discussions in lines 458-466: "It is evident that some CFs are existing at the upper level in the Antarctic in JJA when U_pdf or T_pdf of GTS is used. However, such high CFs are not seen in CloudSat/CALIPSO observations, suggesting that the usage of GTS schemes could cause significant biases in CFs under such environmental conditions. This is of course highly related to the ice CF schemes of GTS. More observation-constrained adjustments or tuning of the ice CF schemes of GTS are needed to reduce the biases in CFs in similar atmospheric environments like

the upper level of the Antarctic winter. Potential tuning parameters of ice CF scheme of GTS are sup and RHc which are discussed in Section 5.7c.".

3.5 Figure 8: I think the scatter plot does not show a great linear correlation between CF and RH, more like exponentials. I am not sure whether I should agree with the author's "higher" linear correlation efficient to evaluate good or bad relationships. Thanks for the comments. You are right. We have removed all the relevant figures both in the main text or in the supplementary material.

3.6 this paper gives out many statistical results for evaluation, but the reasons behind and the connection between modified clouds and change of other related variables are discussed less. I recommend the authors to add more discussion about this. Thank you for the comment. Yes. We have tried to add some more discussions about the reasons behind as possible. Admittedly, it is not easy to disentangle the relationship between causes and effects resulted from the usage of the GTS scheme in the global simulations as we discussed in Section 5.6 of the revised manuscript in lines 702-726: "5.6 Discussions on causality resulted from the GTS scheme Admittedly, it is challenging to disentangle the relationship between causes and effects resulted from the usage of the GTS scheme in the global simulations. As shown in the previous section, utilizing GTS schemes yield changes in cloud fields (both CF and CWC) and cloud radiative forcings. Mean climatology and the annual cycles of many climatic parameters (as discussed in Sections 5.2 and 5.3) are changed to some extent through the cloud–radiation interactions. Notably, such changes in cloud fields are not only contributed by the stratiform cloud macrophysics scheme but also affected by other moist processes in GCMs (e.g., deep convection, shallow convection, stratiform cloud microphysics, and turbulent boundary layer schemes). Moreover, cloud overlapping assumptions in the macrophysics scheme of CESM (both in the horizontal and vertical directions) also affect the global simulation results through changes in thermodynamic and dynamic fields caused by utilizing different cloud macrophysics schemes. We suggest that those asymmetric changes in total CF, SWCF, and LWCF between the tropics

and the mid- and high latitudes (Figure 17) could be related to regions where stratiform cloud macrophysics parameterization takes effect more compared to other moist parameterizations in the physical-process splitting framework of CESM. Such asymmetric changes in cloud radiative forcing are in turn likely to affect the climate mean state and atmospheric circulation. More so-called process-oriented analyses and simulation designs can be devoted to unveiling the causality resulted from the GTS scheme. For example, detailed output of tendency terms of moist processes of GCMs can be useful to investigate how individual moist process responses to the perturbations caused by the GTS scheme and interact together to produce those different changes in low, middle and high clouds.".

3.7 How sensitive the final model's performance to the tuning parameter – sup? From Figure S4, this parameter could reduce the cloud fraction up to 20%, and this will bring a substantial effect on radiation. How the overall model performance will be? Maybe a Taylor plot with different sensitivity simulations could give a good evaluation. Thank you for the suggestions. Yes. We have added relevant discussions in lines 799-812: "As mentioned in the previous subsection, the sup value can be tuned and CF profiles would be modified accordingly as shown in Figure S5. It is thus worth discussing the sensitivity of tuning parameters of the GTS scheme and whether such tuning would affect overall model performance. It is interesting to note that, although significant changes in CF profiles (Figure S5), SWCF, and LWCF (Table S8) between sup = 1.0 and sup = 1.05 are shown, differences in net radiation at the top of model (RESTOM) between sup = 1.0 and sup = 1.05 are only about 0.6 to 0.7 W m-2 for the GTS schemes (Table S8). Such outcome suggests that possible compensating effects exist between changes in SWCF and LWCF associated with cloud overlapping. One could expect that, despite relatively smaller changes in RESTOM, significant changes in SWCF and LWCF between sup = 1.0 and sup = 1.05 could potentially affect the overall performance of GCMs. Comparisons of Taylor diagrams and biases confirm this (Figures S6 and S7, Table S9). Notably, sup here is assumed to be a constant and height-independent. Further height-dependent tuning can be tested.".

Minor comments: 1. Figure 2: U_pdf performs better at a lower level and has a similar performance with T_pdf at the upper level. what might be the potential reason? Thanks for the comments. We have added discussions in lines 350-355: "It is interesting to note that the U_pdf simulates CFs at the lower levels in closer agreement with those of ERA-Interim and the U_pdf obtains similar magnitude of CFs as those of the T_pdf at the upper levels. The potential reason resulted in such differences could be related to the nature of the two PDFs. The U_pdf is likely to calculate more CFs compared to T_pdf given similar RH and cloud liquid mixing ratio in the lower atmospheric levels.".

2. L263: in section 3.1, please include the references for each observation data you used and the time period used. Thanks for the suggestions. Yes. We have provided the relevant information in the revised manuscript.

3. L277-279: do you use another CF and CWC data here? It is slightly confusing. I suggest reorganizing this paragraph. Thanks for pointing out this concern. We have clarified this by rephrasing this paragraph as in lines 285-288: "This dataset (provided by the AMWG diagnostics package of NCAR) is used to compare with CF simulated by the COSP satellite simulator of CESM 1.2.2. Notably, this dataset is different from the one below which also includes cloud water content (CWC).".

4. L 279: "The methodology from Li et al. (2012) is used to generate gridded data". Are there any specifications for this method to generate gridded data? It might be better to briefly describe the method used here and the reader would not take more time to read the referred paper to figure out what the method is. Thank you for the suggestions. Yes. We have added this in lines 306-309: "Two independent approaches (i.e., FLAG and PSD methods) are used in Li et al. [2012] to distinguish ice mass associated with clouds from ice mass associated with precipitation and convection. The PSD method is used in this study [Chen et al., 2011].".

5. L388-390: "on the other hand, . . ..." This sentence is confusing. Please consider rephrasing it. Yes. We have rephrased this sentence in lines 448-450: " On the other

hand, the results of the Park scheme show clouds at higher altitudes in the tropics in closer agreement with CloudSat/CALIPSO than those of U_pdf or T_pdf.".

6. Figure 17: what contributes to the stronger DTCOND in U_pdf compared to T_pdf below 700 hPa? The authors might want to add more discussions about it. Thanks for the comment. Yes. We have added more discussions in lines 647-652: "It is also evident that DTCOND simulated by the U_pdf is stronger than that simulated by the T_pdf below 700 hPa. Such enhanced condensation heating is probably contributed by the enhanced shallow convection as a result of changes in cloud-radiation inter-actions. However, more process-oriented diagnostics are needed to understand the complicated interactions of the moist processes.".

Technical: 1. Figures: labels in some figures are too small. Please increase the font size. Yes. We have increased the font size of labels which are too small in several figures (i.e., Figures 6, 8, 9, 16).

2. Figure 9, the lat-lon plot is also too small. Please consider another type of layout for this figure. Thanks for noticing this issue. We have moved the lat-lon plots to the supplementary material and enlarged these plots.

3. Figure 17: add the unit of each variable Yes. We have added the unit of each variable in the figure caption.

4. Please consider add legends for line plots. Yes. We have added the legends in the line plots.

5. Figure S4: if my understanding is correct, the red solid lines in the upper panel and the lower panel should be from the default Park. However, it looks like they are not the same. Please check it. Thank you for pointing out such inconsistency. We have checked the red solid lines and noticed that the red lines in the upper and lower panel are from the U_pdf with Slingo cloud ice CF parameterization and the T_pdf with Slingo cloud ice CF parameterization, respectively. We have corrected this in the

figure caption.

Please also note the supplement to this comment:
https://gmd.copernicus.org/preprints/gmd-2020-144/gmd-2020-144-AC1-
supplement.pdf

———————————————————

---

## Author Response (AR2)

Author reply:

We would like to thank the anonymous reviewer for the useful comments, which have greatly helped us to improve the manuscript. Here is the point-by-point reply to reviewer's comments. In the following, the texts with italic font are the reviewer's original comments, and the texts with normal font are authors' responses. The line numbers refer to the clean (no track) version of the revised manuscript.

*Anonymous Referee #2*
*The 2nd round of reviewing 'GTS v1.0: A Macrophysics Scheme for Climate Models Based on a Probability Density Function'*

*The authors make efforts to address my comments and the revised manuscript is improved substantially. Overall it is well written and the results are comprehensive. I think this work has important implications for model development. I recommend the paper to be accepted for publication after addressing the following minor comments:*

Thank you very much for the useful comments. Please kindly find our responses as listed below.

*L298-299: this link does not work.*

Thanks for noticing this issue. Yes, we have revised the link and now it works. Please find the corrected link in line 298.

*L324: "410in"?*

Thanks. We have corrected it as shown in line 324.

*L597: "Table 8" -- don't have it now. Please update it.*

Thanks. We have updated it as "Table S6" as shown in line 596.

*maybe the authors could consider moving section 5.6 to the final section with a concise discussion.*

Thank you for the suggestion. Yes. We have moved it to the final section and revised it to become more concise as the suggestion. Please find the revision in lines 860-874: "Admittedly, it is challenging to disentangle the relationship between causes and effects resulted from the usage of the GTS scheme in the global simulations. Notably, such changes in cloud fields and cloud radiative forcings are not only contributed by the stratiform cloud macrophysics scheme but also affected by other moist processes in GCMs (e.g., deep convection, shallow convection, stratiform cloud microphysics, and turbulent boundary layer schemes). Moreover, cloud overlapping assumptions in the macrophysics scheme of CESM (both in the horizontal and vertical directions) also affect the global simulation results through changes in thermodynamic and dynamic fields caused by utilizing different cloud macrophysics schemes. We suggest that those asymmetric changes in total CF, SWCF, and LWCF between the tropics and the mid- and high latitudes could be related to regions where stratiform cloud macrophysics parameterization takes effect more compared to other moist parameterizations in the physical-process splitting framework of CESM. More so-called process-oriented analyses and simulation designs can be devoted to unveiling the causality resulted from the GTS scheme."

*tick labels are still too small in Figure 2.*

Thank you for the suggestion. Yes. We have replotted Figure 2 and enlarged those tick labels.

*please add the meaning of those numbers in the figure caption.*

[revised manuscript text omitted]